# Antagonism of PP2A is an independent and conserved function of HIV-1 Vif and causes cell cycle arrest

Sara Marelli[1,2], James C Williamson[1,2], Anna V Protasio[1,2], Adi Naamati[1,2], Edward JD Greenwood[1,2], Janet E Deane[3,4], Paul J Lehner[1,2], Nicholas J Matheson[1,2]*

[1]Department of Medicine, University of Cambridge, Cambridge, United Kingdom; [2]Cambridge Institute of Therapeutic Immunology and Infectious Disease (CITIID), University of Cambridge, Cambridge, United Kingdom; [3]Department of Clinical Neuroscience, University of Cambridge, Cambridge, United Kingdom; [4]Cambridge Institute for Medical Research (CIMR), University of Cambridge, Cambridge, United Kingdom

**Abstract** The seminal description of the cellular restriction factor APOBEC3G and its antagonism by HIV-1 Vif has underpinned two decades of research on the host-virus interaction. We recently reported that HIV-1 Vif is also able to degrade the PPP2R5 family of regulatory subunits of key cellular phosphatase PP2A (PPP2R5A-E; Greenwood et al., 2016; Naamati et al., 2019). We now identify amino acid polymorphisms at positions 31 and 128 of HIV-1 Vif which selectively regulate the degradation of PPP2R5 family proteins. These residues covary across HIV-1 viruses in vivo, favouring depletion of PPP2R5A-E. Through analysis of point mutants and naturally occurring Vif variants, we further show that degradation of PPP2R5 family subunits is both necessary and sufficient for Vif-dependent G2/M cell cycle arrest. Antagonism of PP2A by HIV-1 Vif is therefore independent of APOBEC3 family proteins, and regulates cell cycle progression in HIV-infected cells.

*For correspondence:
njm25@cam.ac.uk

Competing interests: The authors declare that no competing interests exist.

## Introduction

The canonical function of HIV-1 Vif is to recruit the cellular restriction factor APOBEC3G for CUL5 E3 ligase and ubiquitin-proteasome-dependent degradation in infected cells, preventing APOBEC3G encapsidation and enhancing virion infectivity (*Conticello et al., 2003*; *Kobayashi et al., 2005*; *Marin et al., 2003*; *Mehle et al., 2004*; *Sheehy et al., 2002*; *Sheehy et al., 2003*; *Stopak et al., 2003*; *Yu et al., 2003*). This interaction is very likely to be important in vivo, because the ability of Vif to antagonise APOBEC3G and its homologues is broadly conserved across lentiviral phylogeny, and has driven co-evolution of the mammalian APOBEC3 family (*Compton et al., 2013*; *Nakano et al., 2017*).

The other cell biological phenotype associated with Vif in multiple studies is the induction of G2/M cell cycle arrest (*DeHart et al., 2008*; *Evans et al., 2018*; *Izumi et al., 2010*; *Sakai et al., 2006*; *Wang et al., 2007*; *Zhao et al., 2015*). Vif-dependent cell cycle arrest does not require expression of APOBEC3 family proteins, but is reliant on lysine-48 ubiquitination and the same CUL5 E3 ligase complex recruited by Vif to deplete APOBEC3G (*DeHart et al., 2008*). It has therefore been suspected to reflect ubiquitination and degradation of an unknown cellular factor involved in cell cycle progression (*DeHart et al., 2008*). Why only certain HIV-1 Vif variants mediate this effect (*Evans et al., 2018*; *Zhao et al., 2015*), and how widely conserved it is across the lentiviral lineage, have remained unclear.

We have recently discovered that, in addition to APOBEC3 family proteins, Vif is also able to degrade the B56 family of regulatory subunits of the ubiquitous heterotrimeric serine-threonine phosphatase PP2A (PPP2R5A-E) in HIV-1-infected CEM-T4 T cells (*Greenwood et al., 2016*) and primary human CD4+ T cells (*Naamati et al., 2019*). Together with PP1, PP2A is one of the major cellular serine-threonine phosphatases (*Nasa and Kettenbach, 2018*). Specificity of the PP2A holoenzyme is determined by the bound regulatory subunit, and targets of PP2A-B56 have been implicated in a host of cellular processes, including the regulation of mitotic kinases and cell cycle progression (*Foley et al., 2011*; *Grallert et al., 2015*; *Vallardi et al., 2019*; *Yang and Phiel, 2010*).

The ability of Vif to degrade PPP2R5 subunits (illustrated for PPP2R5D in *Figure 1A*) is shared by Vif variants from diverse primate and non-primate lentiviruses (*Greenwood et al., 2016*), suggesting a beneficial effect on viral replication in vivo. In theory, however, depletion of PPP2R5A-E could be dependent on or secondary to the phylogenetically conserved ability of Vif to antagonise APOBEC3 family proteins. To demonstrate that these functions are autonomous and have therefore been independently selected, we now screen a library of rationally designed Vif point mutants, and identify amino acid substitutions at residues 31 and 128 which clearly separate APOBEC3 and PPP2R5 family protein depletion. We further show that antagonism of PP2A explains the ability of Vif to cause cell cycle arrest, and that this requires efficient depletion of all PPP2R5 family subunits. Naturally occurring polymorphisms of residues 31 and 128 correlate with the ability of HIV-1 Vif variants to cause cell cycle arrest, and reveal evidence of selection pressure for PPP2R5A-E depletion in vivo.

## Results

### Flow cytometric screen identifies mutations in HIV-1 Vif which separate PPP2R5B and APOBEC3G depletion

To determine whether antagonism of PPP2R5 and APOBEC3 family proteins are independent functions of Vif, we first used the published structure of the Vif-CUL5 complex (*Guo et al., 2014*) to construct a library of 34 HIV-1 NL4-3 Vif variants with point mutations in solvent-exposed residues, focussing predominantly on regions distant from known APOBEC3 family protein interaction interfaces (*Figure 1B* and *Figure 1—figure supplement 1A*, residues highlighted in yellow). None of these mutations is predicted to cause protein misfolding, nor interfere with the interactions between Vif and other members of the Vif-CUL5 E3 ligase complex (CBF-β, CUL5, ELOB and ELOC).

Amongst the five PPP2R5 family subunits, we previously showed that depletion of PPP2R5B is most conserved across Vif variants from HIV-1/2 and the non-human primate lentiviruses (*Greenwood et al., 2016*). We therefore transfected our library into HEK 293T cells (293Ts) stably expressing HA-tagged PPP2R5B or APOBEC3G, and used flow cytometry to quantify PPP2R5B and APOBEC3G depletion by each Vif variant (*Figure 1—figure supplement 1B–C* and *Figure 1—figure supplement 2A–C*). As well as indicating preserved APOBEC3 family substrate recruitment, the ability to deplete APOBEC3G served as a control for unanticipated effects on Vif expression or stability, or assembly of the Vif-CUL5 complex.

We discovered several Vif mutants to be defective for PPP2R5B depletion (*Figure 1C*). Vif protein expression levels were similar (*Figure 1—figure supplement 3*), but some mutations affected residues already known to be required for depletion of APOBEC3G (K26, Y44, W70) (*Letko et al., 2015*) or APOBEC3C/F (R15) (*Letko et al., 2015*; *Nakashima et al., 2016*; *Figure 1D* and *Figure 1—figure supplement 2C*). Conversely, Vif variants with mutations in residues Y30/I31, R33/K34 and I128 were defective for PPP2R5B depletion, retained the ability to antagonise APOBEC3G, and had not been implicated in APOBEC3C/F depletion. These residues are grouped in three similarly orientated patches on the Vif surface (*Figure 1B*, residues highlighted in red). Aiming to identify Vif variants specifically defective for PPP2R5 subunit depletion, we therefore focused on mutations in residues I128, I31 and R33/K34 for further evaluation, including representatives from each patch.

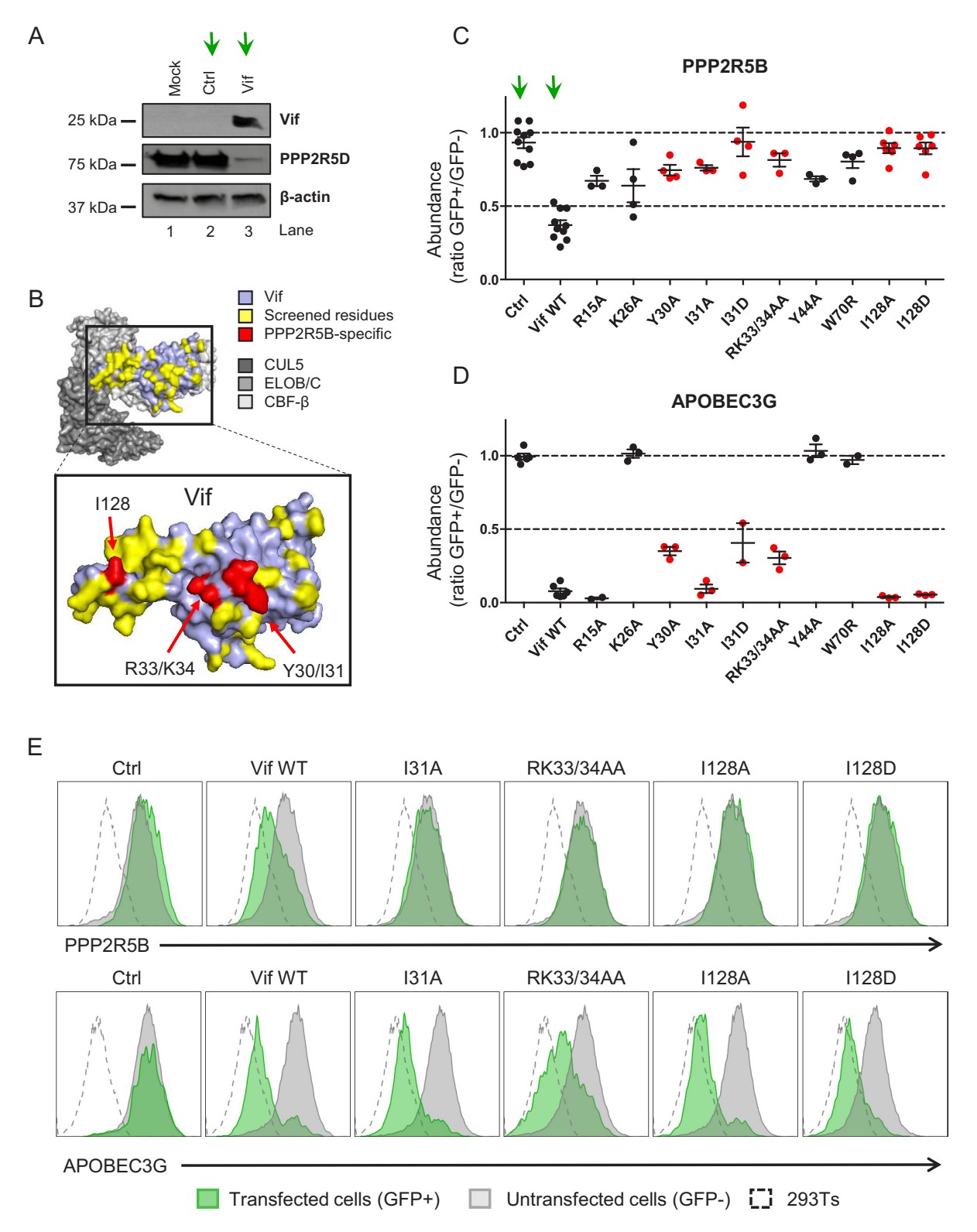

**Figure 1.** Flow cytometric screen of HIV-1 Vif point mutants. (**A**) Depletion of endogenous PPP2R5D by HIV-1 Vif. CEM-T4s were transduced with lentiviruses encoding either EGFP-SBP-ΔLNGFR (Ctrl) or EGFP-P2A-Vif (Vif) at an MOI of 3, then lysed in 2% SDS and analysed by immunoblot with anti-Vif, anti-PPP2R5D and anti-β-actin (loading control) antibodies after 48 hr. Green arrows, Ctrl vs Vif. (**B**) Solvent-accessible surfaces of Vif (pale blue) in complex with CUL5 (dark grey), ELOB/C (grey) and CBF-β (light grey). Residues highlighted in yellow were targeted in our library of point mutants (total

*Figure 1 continued on next page*

Figure 1 continued

34). Residues highlighted in red specifically affected the depletion of PPP2R5B, but not APOBEC3G. (C–D) Depletion of PPP2R5B (C) or APOBEC3G (D) by selected Vif point mutants. 293Ts stably expressing HA-tagged PPP2R5B or APOBEC3G were transfected with constructs encoding EGFP-P2A-Vif, then fixed/permeabilised, stained with AF647-conjugated anti-HA antibody and analysed by flow cytometry after 36 hr (see *Figure 1—figure supplement 1B–C*). For each Vif point mutant, abundance of PPP2R5B or APOBEC3G is shown as a ratio of A4647 fluorescence in GFP+ (transfected, Vif+) to GFP- (untransfected, Vif-) cells, after deducting background fluorescence of control 293Ts (no HA-tagged protein expression). Individual data points represent biological replicates (minimum 3). Mean values with standard error of the mean (SEM) are indicated. Vif point mutants specifically affecting the depletion of PPP2R5B are highlighted in red. Ctrl, control constructs encoding EGFP or EGFP-SBP-ΔLNGFR. Data for other Vif point mutants are shown in *Figure 1—figure supplement 2A–B*. Green arrows, Ctrl vs Vif WT. (E) Representative data from (C–D). Green, GFP+, transfected cells (Vif+); grey, GFP-, untransfected cells (Vif-); dotted line, background staining of control 293Ts (no HA-tagged protein expression).

The online version of this article includes the following figure supplement(s) for figure 1:

**Figure supplement 1.** Further details for site-directed mutagenesis and flow cytometric screen.
**Figure supplement 2.** Complete results of flow cytometric screen.
**Figure supplement 3.** Stability of selected Vif point mutants in 293Ts.
**Figure supplement 4.** Depletion of APOBEC3F by selected Vif point mutants.

## Residues 128 and 31 of HIV-1 Vif differentially regulate APOBEC3 and PPP2R5 family protein depletion

As well as APOBEC3G, other APOBEC3 family members (such as APOBEC3F and APOBEC3H haplotype II) are also able to restrict HIV replication (*Feng et al., 2014*), and Vif recruits different APOBEC3 family members for degradation using distinct binding surfaces (*Binka et al., 2012*; *Chen et al., 2009*; *Dang et al., 2009*; *Gaddis et al., 2003*; *Harris and Anderson, 2016*; *He et al., 2008*; *Letko et al., 2015*; *Mehle et al., 2007*; *Nakashima et al., 2016*; *Ooms et al., 2017*; *Richards et al., 2015*; *Russell and Pathak, 2007*; *Simon et al., 2005*; *Yamashita et al., 2008*). In addition, Vif variants from HIV-1/2 and the non-human primate lentiviruses differ in their abilities to deplete different PPP2R5 family subunits (*Greenwood et al., 2016*). We therefore sought to determine whether the mutations we found to separate depletion of PPP2R5B and APOBEC3G have similar effects on other family members.

First, we tested the ability of Vif mutants lacking the ability to deplete PPP2R5B to deplete HA-tagged APOBEC3F in 293Ts, similar to our initial flow cytometry screen (*Figure 1—figure supplement 4A–C*). As previously reported (*Letko et al., 2015*; *Nakashima et al., 2016*), mutation of R15 resulted in loss of activity against APOBEC3F. The RK33/34AA mutant was also partially impaired, but other mutants retained full activity against APOBEC3F. APOBEC3H haplotype II was not examined, because wildtype NL4-3 Vif (on which our Vif mutant library was based) is unable to deplete this APOBEC3 family member (*Binka et al., 2012*; *Ooms et al., 2013*; *Zhao et al., 2015*).

To avoid the possibility of over-expression artefacts, we next focussed on endogenous APOBEC3 and PPP2R5 family proteins expressed in CEM-T4 T cells (CEM-T4s). These cells were transduced with the panel of I31, I128 and R33/K34 Vif mutants found in our flow cytometry screens to be specifically defective for PPP2R5B depletion, together with a Y44 mutant (also defective for APOBEC3G depletion) to serve as a control.

Levels of an indicative PPP2R5 subunit for which a reliable antibody is available (PPP2R5D, as in *Figure 1A*) were then measured by immunoblot (*Figure 2A*). As expected WT Vif was able to fully deplete PPP2R5D (lane 3). Conversely, mutations in residues I128, I31, R33/K34 and Y44 all restored PPP2R5D levels (lanes 4–9). Interestingly, the I128A (lane 4) and Y44A (lane 9) mutations only showed a partial rescue, suggesting a differential effect on different PPP2R5 subunits (PPP2R5B vs PPP2R5D). In addition, mutations in residues R33/K34 (lanes 7–8) were associated with lower levels of Vif expression (*Figure 2A*).

Validated antibodies capable of detecting and differentiating endogenous levels of all APOBEC3 and PPP2R5 family proteins are not available. We therefore evaluated the activity of a similar panel of Vif mutants using a tandem mass tag (TMT)-based functional proteomic approach (*Figure 2B*). CEM-T4 cells were transduced with different Vif mutants at a multiplicity of infection (MOI) of 3 (range 94.1–98.7% transduced cells), then subjected to whole cell proteome analysis after a further 48 hr.

In total, we identified 8781 proteins (*Figure 2—source data 1*), including all 5 PPP2R5 family subunits (PPP2R5A/B/C/D/E) and 5 out of 7 APOBEC3 family members (B/C/D/F/G; not A/H). This

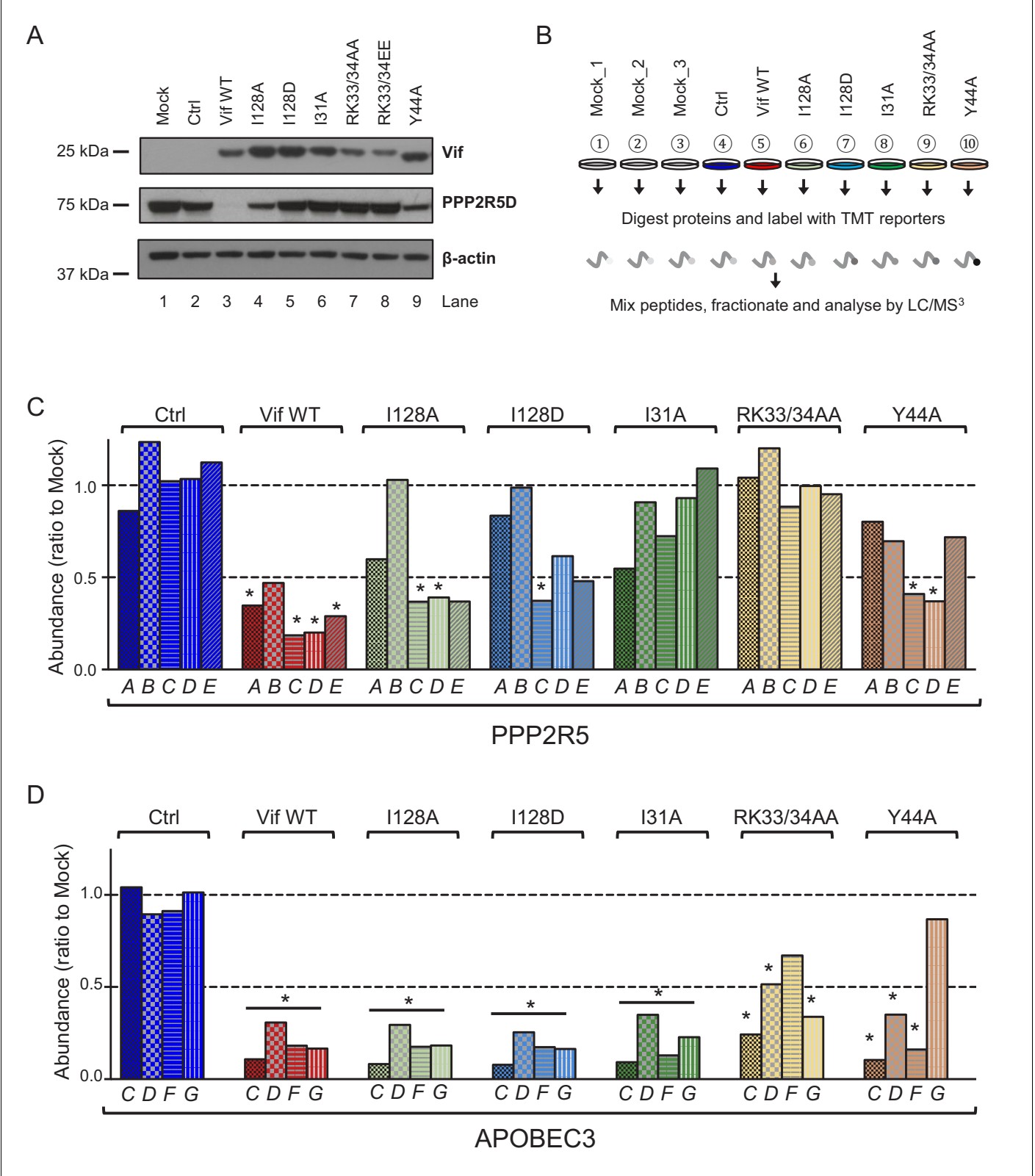

**Figure 2.** Depletion of endogenous APOBEC3 and PPP2R5 family proteins by HIV-1 Vif point mutants. (**A**) Depletion of endogenous PPP2R5D by selected Vif point mutants. CEM-T4s were transduced with lentiviruses encoding EGFP-P2A-Vif at an MOI of 3, then lysed in 2% SDS and analysed by immunoblot with anti-Vif, anti-PPP2R5D and anti-β-actin (loading control) antibodies after 48 hr. Ctrl, control construct encoding EGFP-SBP-ΔLNGFR. (**B**) Overview of proteomic experiment 1 (selected Vif point mutants). CEM-T4s were transduced with lentiviruses encoding EGFP-P2A-Vif at an MOI of 3,

*Figure 2 continued on next page*

Figure 2 continued

then analysed by TMT-based quantitative proteomics after 48 hr. Mock_1/2/3, biological replicates. Ctrl, control construct encoding EGFP-SBP-Δ LNGFR. (C–D) Depletion of endogenous PPP2R5 family (C) or APOBEC3 family (D) proteins by selected Vif point mutants in cells from (B). For each Vif point mutant, abundance of respective PPP2R5 or APOBEC family members is shown as a ratio to the mean abundance of the same family member in the three mock-transduced samples. Significant outliers from the distribution of abundances in mock-transduced samples are highlighted (see Materials and methods and *Figure 2—figure supplement 2* for further details). *p<0.05.

The online version of this article includes the following source data and figure supplement(s) for figure 2:

**Source data 1.** Complete data from proteomic experiment 1 (selected Vif point mutants).
**Figure supplement 1.** Stability of selected Vif point mutants in CEM-T4s.
**Figure supplement 2.** Calculation of *t*-scores and *p*-values.

concords with previous data suggesting that APOBEC3A is restricted to myeloid cells (*Berger et al., 2011*; *Koning et al., 2009*; *Peng et al., 2007*; *Refsland et al., 2010*), and neither APOBEC3A nor APOBEC3H are expressed in CCRF-CEM cell lines at the mRNA level (*Refsland et al., 2010*). APO-BEC3B is not antagonised by Vif (*Doehle et al., 2005*; *Greenwood et al., 2016*; *Hultquist et al., 2011*; *Naamati et al., 2019*), and is therefore not considered further.

As expected, all Vif mutants tested were defective for PPP2R5B depletion, and the Y44A mutant was also defective for APOBEC3G depletion (*Figure 2C–D*). In addition, and consistent with our immunoblot analysis (*Figure 2A*), substitutions of I128 led to loss of activity against PPP2R5A, with relatively preserved activity against PPP2R5C-E. Conversely, substitution of I31 led to a reciprocal pattern, with loss of activity against PPP2R5C-E, but relatively preserved activity against PPP2R5A. Substitutions of R33/K34 led to loss of activity against all PPP2R5 subunits, but were again associated with lower levels of Vif expression (*Figure 2—figure supplement 1*), and accompanied by partial loss of activity against APOBEC3 family proteins, particularly APOBEC3F (mirroring our flow cytometry data, *Figure 1—figure supplement 4A–C*). In conclusion, therefore, mutations in residues 128 and 31 separate PPP2R5 and APOBEC3 family depletion without affecting Vif stability, and differentially regulate the 5 PPP2R5 family members.

## Depletion of PPP2R5 family subunits is necessary for Vif-dependent cell cycle arrest

We previously showed that expression of Vif results in extensive remodelling of the phosphoproteome in HIV-infected cells, including activation of the aurora kinases AURKA and AURKB, effects we attributed to PP2A antagonism (*Greenwood et al., 2016*; *Naamati et al., 2019*). As expected, transduction of CEM-T4s with WT Vif resulted in increased AURKA/B T loop phosphorylation (*Figure 3A*, lane 3). Conversely, Vif mutants with impaired ability to antagonise PPP2R5 family subunits were unable to trigger AURKA/B phosphorylation (*Figure 3A*, lanes 4–7).

Together with APOBEC3 family antagonism, it has been known for >10 years that certain Vif variants (including NL4-3 Vif) are also able to induce G2/M cell cycle arrest, and that this is dependent on CUL5 E3 ligase recruitment and the ubiquitin-proteasome system (*DeHart et al., 2008*; *Evans et al., 2018*; *Izumi et al., 2010*; *Sakai et al., 2006*; *Wang et al., 2007*; *Zhao et al., 2015*). The Vif substrate explaining this phenomenon has, however, remained obscure.

Since both PP2A-B56 (PP2A heterotrimers incorporating one of the B56 family of regulatory subunits, PPP2R5A-E) and aurora kinases are required to coordinate mitotic progression (*Foley et al., 2011*; *Grallert et al., 2015*; *Nasa and Kettenbach, 2018*; *Vallardi et al., 2019*), we hypothesised that depletion of PPP2R5 family subunits may explain Vif-dependent cell cycle arrest, and that Vif mutants with impaired activity against PPP2R5 family subunits may also be defective for this phenotype. To test this hypothesis, we first interrogated our proteomic dataset. As predicted, WT Vif led to elevated levels of cyclin B1, indicative of G2/M arrest (*Figure 3B*). Conversely, elevation of cyclin B1 was reduced or abolished in the presence of Vif mutants lacking the ability to deplete PPP2R5 family subunits.

To confirm this result and formally evaluate cell cycle progression, we measured DNA content of CEM-T4s 48 hr after transduction with WT or mutant Vif variants. Again, WT Vif, but not Vif mutants lacking the ability to deplete PPP2R5 subunits, caused G2/M arrest (*Figure 3C–D*). As a control, two other Vif mutants (F39A and D61A) which retained the ability to antagonise PPP2RB (*Figure 1—*

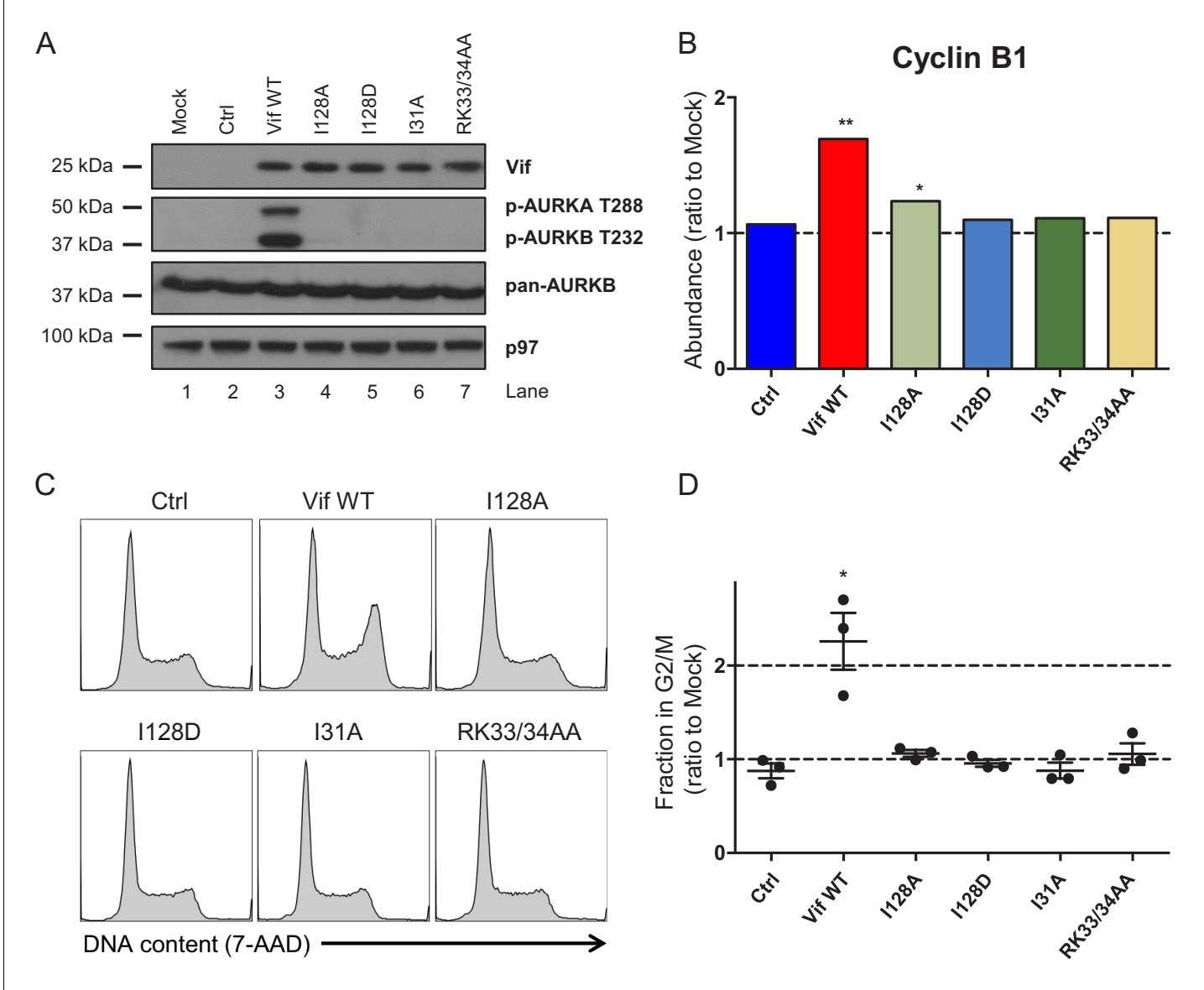

**Figure 3.** Regulation of cell cycle by HIV-1 Vif point mutants. (**A**) Phosphorylation of aurora kinases in the presence of selected Vif point mutants. CEM-T4s were transduced with lentiviruses encoding EGFP-P2A-Vif at an MOI of 3, then lysed in 2% SDS and analysed by immunoblot with anti-Vif, anti-phospho-AURK, anti-total AURKB and anti-p97 (loading control) antibodies after 48 hr. Ctrl, control construct encoding EGFP-SBP-ΔLNGFR. (**B**) Regulation of cyclin B1 by selected Vif point mutants in cells from proteomic experiment 1 (*Figure 2B*). For each Vif point mutant, abundance of cyclin B1 is shown as a ratio to the mean abundance in the three mock-transduced samples. Significant outliers from the distribution of abundances in mock-transduced samples are highlighted (see Materials and methods for details). *p<0.05; **p<0.05. (**C–D**) Regulation of cell cycle by selected Vif point mutants. CEM-T4s were transduced with lentiviruses encoding EGFP-P2A-Vif at an MOI of 3, then fixed in 90% methanol, stained with 7-AAD and analysed by flow cytometry after 48 hr. Representative data (**C**) from three biological replicates (**D**) are shown. For each Vif point mutant, the fraction of cells in G2/M is shown as a ratio to the fraction of cells in G2/M in mock-transduced cells. Individual data points reflect biological replicates. Mean values with SEM are indicated. Significant differences compared with mock-transduced cells are highlighted (*t*-tests). *p<0.05. Ctrl, control construct encoding EGFP-SBP-ΔLNGFR.

The online version of this article includes the following figure supplement(s) for figure 3:

**Figure supplement 1.** Additional controls for cell cycle analysis (Vif point mutants).

**Figure supplement 2.** Depletion of endogenous DPH7 and FMR1 by selected Vif point mutants.

*figure supplement 2A*), also retained the ability to cause cell cycle arrest (*Figure 3—figure supplement 1*).

In addition to APOBEC3 and PPP2R5 family proteins, we recently showed that NL4-3 Vif is also able to target FMR1 and DPH7 (*Naamati et al., 2019*). Both I128A and I128D point mutants retain the ability to deplete these proteins (*Figure 3—figure supplement 2*), but are unable to mediate cell cycle arrest. Depletion of PPP2R5 family subunits, but not other Vif substrates, is therefore required for Vif-dependent aurora kinase activation and G2/M cell cycle arrest.

## Depletion of PPP2R5 family subunits is sufficient to cause cell cycle arrest

Consistent with PPP2R5A-E depletion by Vif, inhibition of PP2A with okadaic acid causes G2/M cell cycle arrest in CEM-T4s (*Figure 4—figure supplement 1A*). However, whilst relatively specific for PP2A over other cellular phosphatases, okadaic acid does not distinguish individual PPP2R5 family subunits, nor separate PP2A-B56 activity from the activity of other PP2A heterotrimers incorporating regulatory subunits from different families (*Swingle et al., 2007*).

Since Vif-dependent cell cycle arrest is abrogated by point mutations which rescue quite distinct PPP2R5 subunits (compare *Figure 2C* and *Figure 3D*), some functional redundancy between the different B56 family members seems likely. Indeed, all PPP2R5 family subunits share a well conserved substrate-binding pocket (*Hertz et al., 2016*; *Wang et al., 2016*), and previous studies have suggested functional equivalence in mitosis (*Foley et al., 2011*; *Lee et al., 2017*). Conversely, another more recent study suggested topological restriction of PPP2R5 subunit activity within cells (*Vallardi et al., 2019*). We therefore sought to test the requirement for different PPP2R5 subunits for cell cycle progression using combinatorial knockdowns. To permit this approach, we used HeLa cells (HeLas) as a model system.

First, we confirmed that, as in CEM-T4s, expression of WT NL4-3 Vif in HeLas causes cell cycle arrest (*Figure 4—figure supplement 1B*). Likewise, mRNA expression levels of individual PPP2R5 subunits were determined by quantitative real-time PCR (qRT-PCR), and found to be similar between cell types, with PPP2R5B much lower than other subunits (*Figure 4—figure supplement 2A*). Next, we transfected HeLas with siRNA targeting individual PPP2R5 subunits, or a pool of siRNA simultaneously targeting all subunits (same total siRNA concentration). Strikingly, we only observed cell cycle arrest when all subunits were knocked down together (*Figure 4A–B*, 'pool'). Indistinguishable results were seen for two independent panels of PPP2R5 family subunit siRNAs, and efficiency of siRNA knockdown was confirmed by qRT-PCR (*Figure 4—figure supplement 2B*).

To confirm that knockdown of all PPP2R5 subunits is necessary for cell cycle arrest, we repeated the experiment using pools of siRNA targeting 4 out of 5 PPP2R5 subunits ('minus one'). Again, and with the exception of PPP2R5B, cell cycle arrest was only observed when all subunits were knocked down together (*Figure 4C–D*). Interestingly, near-identical results were previously reported from RPE1 cells (*Lee et al., 2017*).

That depletion of PPP2R5B is neither sufficient (*Figure 4A–B*) nor required (*Figure 4C–D*) in this setting may reflect low expression of PPP2R5B (*Figure 4—figure supplement 2A*), consistent with previous protein-level data from HeLas (*Geiger et al., 2012*). To test this hypothesis, we generated HeLa cells expressing exogenous PPP2R5B at similar levels to other PPP2R5 subunits (*Figure 4—figure supplement 2C*). As predicted, transfection of a pool of siRNA targeting PPP2R5A, C D and E (but not B) caused cell cycle arrest in wildtype HeLas, but not HeLas over-expressing PPP2R5B (*Figure 4—figure supplement 2D–E*, compare with *Figure 4C–D*).

Finally, knockdown of FMR1 and DPH7 (*Figure 4—figure supplement 3*) did not cause cell cycle arrest. Taken together, these observations therefore explain why efficient depletion of all PPP2R5 subunits is required to cause cell cycle arrest, and why Vif variants with impaired activity against any PPP2R5 subunit are defective for this phenotype.

## Naturally occurring vif variants phenocopy I31 and I128 point mutants

The ability to cause G2/M cell cycle arrest is known to vary between naturally occurring HIV-1 Vif variants from clade B viruses (such as NL4-3 and HXB2), as well as viruses from other clades (*Evans et al., 2018*; *Zhao et al., 2015*). We therefore examined conservation of residues 31, 33/34 and 128 across 2171 clade B HIV-1 Vif sequences available from the Los Alamos National Laboratory

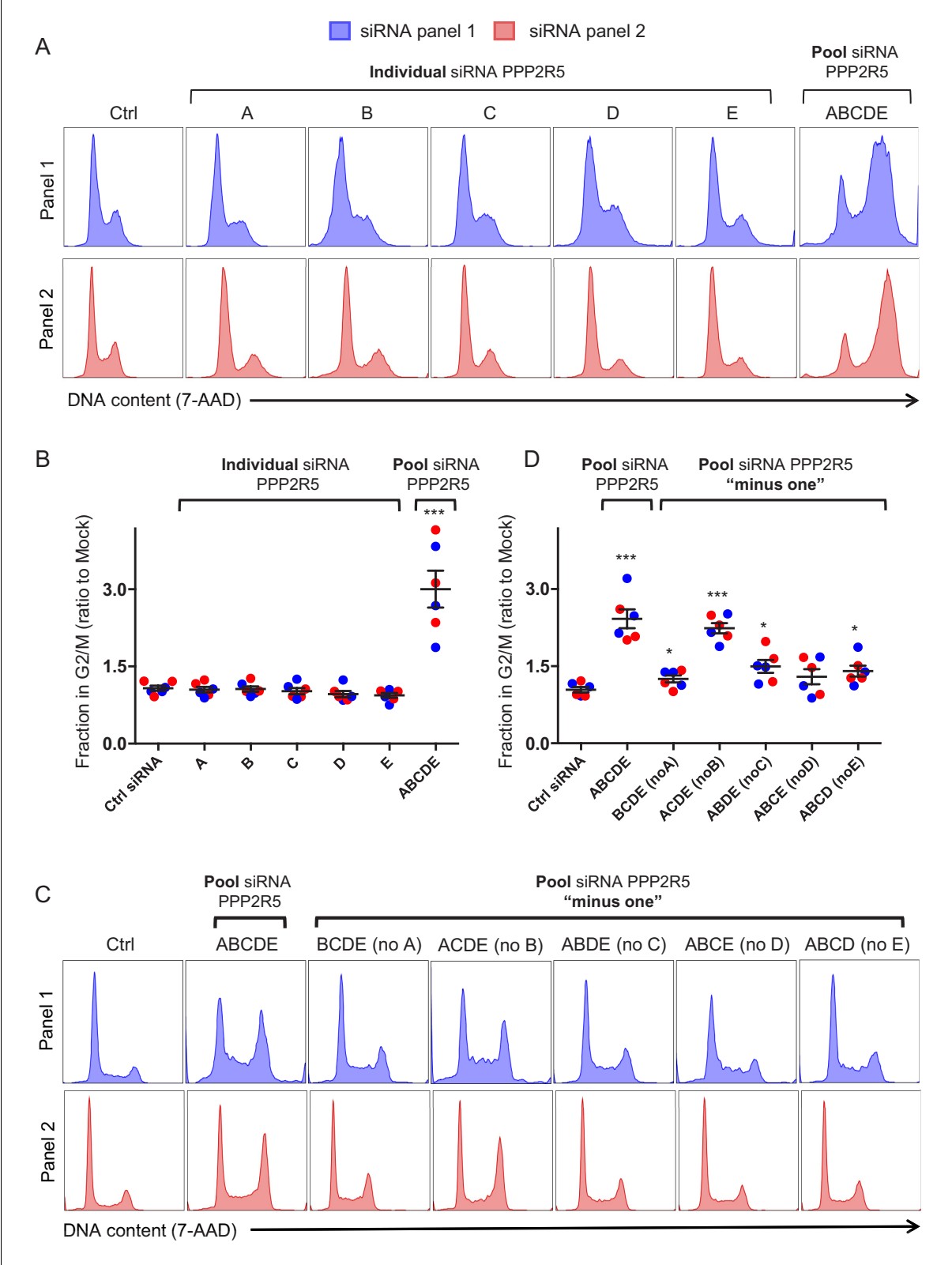

**Figure 4.** Regulation of cell cycle by depletion of PPP2R5 family subunits. (A–B) Regulation of cell cycle by individual vs pooled PPP2R5A-E siRNA. HeLas were transfected with the indicated siRNA, then fixed in 90% methanol, stained with 7-AAD and analysed by flow cytometry after 48 hr. Representative data (A) from three biological replicates (B) for each of 2 panels of siRNA are shown. For each condition, the fraction of cells in G2/M is shown as a ratio to the fraction of cells in G2/M in mock-transfected cells. Individual data points reflect biological replicates. Mean values with SEM are

*Figure 4 continued on next page*

Figure 4 continued

indicated. Significant differences compared with mock-transduced cells are highlighted (*t*-tests). *p<0.05. ***p<0.0005. Ctrl siRNA, MISSION siRNA Universal Negative Control #1. Blue histograms/data points, siRNA panel 1. Red histograms/data points, siRNA panel 2. (C–D) Regulation of cell cycle by combinations of pooled PPP2R5A-E siRNA. HeLas were transfected with the indicated siRNA, then fixed in 90% methanol, stained with 7-AAD and analysed by flow cytometry after 48 hr. Representative data (C) from three biological replicates (D) for each of 2 panels of siRNA are shown. All details as per (A–B).

The online version of this article includes the following figure supplement(s) for figure 4:

**Figure supplement 1.** Additional controls for cell cycle analysis (okadaic acid and Vif).
**Figure supplement 2.** Additional controls for cell cycle analysis (PPP2R5A-E siRNA).
**Figure supplement 3.** Additional controls for cell cycle analysis (DPH7 and FMR1 siRNA).

Web Alignments database (*Figure 5A*). Similar results were obtained when 3412 Vif sequences from all (any clade, including B) non-recombinant HIV-1 M group viruses were considered (*Figure 5—figure supplement 1*).

Interestingly, residues 31 (I or V), 33 (K, G or R) and 128 (I or R, or less commonly L or V) all showed obvious polymorphism, with NL4-3 Vif encoding the commonest amino acids at positions 31 (I) and 128 (I), and the second commonest amino acid at position 33 (R). We therefore evaluated each of the common polymorphisms as single point mutations on a background of NL4-3 Vif using our flow cytometric screen. Conservative substitutions in positions 31 and 128 partially impaired the ability of NL4-3 Vif to deplete PPP2R5B (I31V, I128V and I128L), whereas I128R resulted in more marked impairment (*Figure 5B*). Likewise, R33K, but not R33G, was well tolerated.

To evaluate these polymorphisms in their natural context, we tested Vif variants from two further clade B HIV-1 viruses in our flow cytometric assay: the HIV-1 reference strain HXB2 (encoding 31V, 33G and 128I), and the macrophage-tropic patient isolate YU2 (encoding 31I, 33G and 128R). As a control, we also included a Vif variant from the clade B transmitted founder virus CH470 (encoding 31I, 33K and 128I, similar to NL4-3 Vif) (*Figure 5C*). Consistent with the observed substitutions at residues 31, 33 and 128, HXB2 and YU2 Vif variants were markedly impaired for PPP2R5B depletion, whereas CH470 Vif was at least as active as NL4-3 Vif (*Figure 5D*). Depletion of APOBEC3G was preserved in each case (*Figure 5—figure supplement 2*).

To further assess the function of these variants against other APOBEC3 and PPP2R5 family members, we again adopted a TMT-based functional proteomic approach (*Figure 6A*). As well as HXB2, YU2 and CH470 Vif variants, we included NL4-3 Vif variants with corresponding point mutations at positions 31 and 128 (I31V and I128R). In practice, since residue 127 is also polymorphic, and residues 127 and 128 together overlap a critical HIV splicing silencer (*Madsen and Stoltzfus, 2005*), we combined I128R and R127Y mutations (RI127/128YR, as found in YU2 Vif and detailed in *Figure 6—figure supplement 1*). Finally, to test the combinatorial effect of mutations in residues 31 and 128, we included an NL4-3 Vif variant encoding both I31A and RI127/128YR (Vif AYR). CEM-T4 cells were transduced with the panel of Vif variants at an MOI of 3 (range 93.9–98.4% transduced cells), then subjected to whole cell proteome analysis after a further 48 hr.

In this experiment, we identified 8265 proteins (*Figure 6—source data 1*), including 4 out of 5 PPP2R5 family subunits (A/C/D/E) and 4 out of 7 APOBEC3 family members (B/C/D/G). As expected, CH470 Vif remained fully active against all PPP2R5 (*Figure 6B*) and APOBEC3 (*Figure 6C*) family subunits. Conversely, both YU2 and (in particular) HXB2 Vif variants were selectively impaired for PPP2R5 depletion (*Figure 6B*). In each case, the differential pattern of PPP2R5 family subunit depletion mirrored the effects of corresponding point mutations of residue 31 (HXB2 and NL4-3 I31V, mainly affecting PPP2R5C-E) or 128 (YU2 and NL4-3 RI127/128YR, mainly affecting PPP2R5A). Interestingly, whilst qualitatively similar to the I31A, I128A and I128D Vif mutants evaluated earlier, Vif variants with these naturally occurring polymorphisms were less severely impaired (compare *Figure 2C* with *Figure 6B*).

As a functional readout, we tested the ability of the same panel of Vif variants to cause cell cycle arrest. Consistent with previous reports (*Evans et al., 2018*; *Zhao et al., 2015*) and correlating with their impaired activity against PPP2R5 family subunits, HXB2 and NL4-3 I31V Vif variants were unable to cause cell cycle arrest (*Figure 6D* and *Figure 6—figure supplement 2*). Similarly, the potency of YU2 and NL4-3 RI127/128YR Vif variants was greatly reduced, but not abolished. Naturally occurring

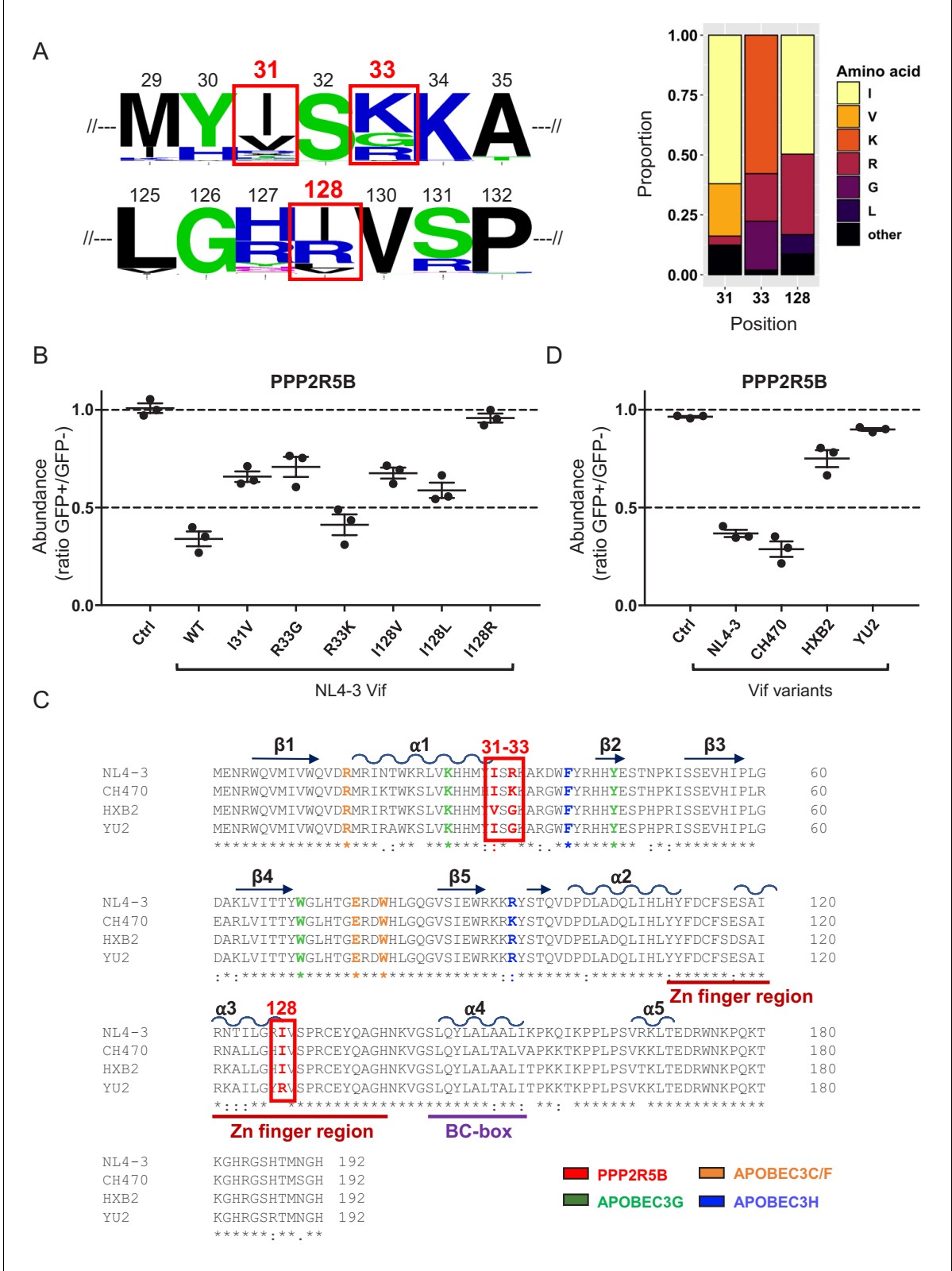

**Figure 5.** Analysis of naturally occurring HIV-1 Vif variants. (**A**) Amino acid polymorphism amongst 2171 naturally occurring HIV-1 M group Vif variants (clade B). Sequence logos (left panel) and bar chart (right panel) highlight frequencies of amino acids corresponding to residues 31, 33 and 128 of NL4-3 Vif. In the sequence logos, polar amino acids (AAs) are depicted in green; neutral AAs, in purple; basic AAs, in blue; acidic AAs, in red; and hydrophobic AAs, in black. An equivalent bar chart for all naturally occurring non-recombinant HIV-1 M group Vif variants (all clades) is shown in

*Figure 5 continued on next page*

*Figure 5 continued*

*Figure 5—figure supplement 1*. (B) Depletion of PPP2R5B by selected Vif point mutants. 293Ts stably expressing HA-tagged PPP2R5B were transfected with constructs encoding EGFP-P2A-Vif, then fixed/permeabilised, stained with AF647-conjugated anti-HA antibody and analysed by flow cytometry after 36 hr. Ctrl, control construct encoding EGFP-SBP-ΔLNGFR. All details as per *Figure 1C*. (C) Sequence alignments of selected Vif variants. Amino acids corresponding to residues 31, 33 and 128 of NL4-3 Vif are highlighted in red (red boxes). Other residues targeted in our library of point mutants and known to interact with APOBEC3G (green), APOBEC3C/F (orange) and APOBEC3H (blue) are also shown (as per *Figure 1—figure supplement 2C*). Additional annotations (α-helices, β-sheets, Zn finger and BC-box) are based on the published Vif-CUL5 crystal structure (*Guo et al., 2014*). (D) Depletion of PPP2R5B by selected Vif variants. 293Ts stably expressing HA-tagged PPP2R5B were transfected with constructs encoding EGFP-P2A-Vif, then fixed/permeabilised, stained with AF647-conjugated anti-HA antibody and analysed by flow cytometry after 36 hr. Ctrl, control construct encoding EGFP-SBP-ΔLNGFR. All details as per (B) and *Figure 1C*.

The online version of this article includes the following figure supplement(s) for figure 5:

**Figure supplement 1.** Amino acid polymorphism amongst 3412 naturally occurring non-recombinant HIV-1 M group Vif variants (all clades).

**Figure supplement 2.** Depletion of APOBEC3G by selected Vif variants.

polymorphisms at residues 31 and 128 therefore modulate the ability of Vif to deplete PPP2R5 family subunits, and explain why some HIV-1 Vif variants are unable to cause cell cycle arrest.

## Combined I31 and I128 mutations abolish PPP2R5 family subunit depletion and rarely occur in nature

Unlike individual mutations of residues 31 or 128, combined I31A and RI127/128YR mutations in NL4-3 Vif (Vif AYR) completely abolished the depletion of all PPP2R5 family subunits in our proteomic analysis (*Figure 6B*), without affecting the depletion of APOBEC3 family proteins (*Figure 6C*). Since PPP2R5B and APOBEC3F were not quantitated in the proteomic experiment, we tested the ability of Vif AYR to deplete these subunits by flow cytometry in 293Ts, including APOBE3G as a control (*Figure 6—figure supplement 3A–B*). As expected, Vif AYR was able to deplete APOBEC3F and APOBEC3G but not PPP2R5B, and failed to cause cell cycle arrest (*Figure 6D*).

We next sought to confirm these results in the context of viral infection by introducing the same mutations into the NL4-3-based HIV-AFMACS molecular clone (*Naamati et al., 2019*). This Env-deficient (single round) reporter virus encodes a SBP-ΔLNGFR cell surface streptavidin-binding affinity tag, allowing facile one-step selection of infected cells with streptavidin-conjugated magnetic beads (Antibody-Free Magnetic Cell Sorting, AFMACS) (*Matheson et al., 2014*). To enable analysis of cell cycle without confounding by Vpr, a Vpr-deficient (ΔVpr) background was used.

To assess the function of Vif AYR against APOBEC3 and PPP2R5 family members during viral infection, we first adopted a TMT-based functional proteomic approach to compare mock-infected cells with cells infected with ΔVpr-Vif WT, ΔVpr-ΔVif or ΔVpr-Vif AYR viruses (*Figure 7A*). CEM-T4s were infected at an MOI of 0.5 (range 29.3–48.7% infected cells), purified using AFMACS after 48 hr (range 93.3–96.6% infected cells, *Figure 7—figure supplement 1A–B*), then subjected to whole cell proteome analysis.

In this experiment, we identified 6297 proteins (*Figure 7—source data 1*), including 4 out of 5 PPP2R5 family subunits (A/C/D/E) but only 1 out of 7 APOBEC3 family member (C). As expected, ΔVpr-Vif WT virus (*Figure 7B*, left panel) but not ΔVpr-ΔVif virus (*Figure 7B*, middle panel) was able to deplete both APOBEC3C and PPP2R5 family proteins. Conversely, ΔVpr-Vif AYR virus (*Figure 7B*, right panel) retained the ability to deplete APOBEC3C, but was completely inactive against PPP2R5 family proteins. As a control, the Nef and Vpu target CD4 was similarly downregulated by each virus (*Guy et al., 1987*; *Willey et al., 1992*). To confirm a functional effect on PP2A, we then used these viruses to infect CEM-T4 T cells, and measured their effect on cell cycle progression. Again, only ΔVpr-Vif WT virus, but not ΔVpr-ΔVif or ΔVpr-Vif AYR viruses, was able to induce G2/M cell cycle arrest (*Figure 7C*). A similar effect was observed in HIV-infected primary human CD4+ T cells (*Figure 7—figure supplement 2A–B*).

If the ability of Vif to antagonise PP2A is maintained by selection pressure in vivo, combinations of unfavourable (less active against PPP2R5A-E) mutations in residues 31 and 128 (abolishing all PPP2R5 family subunit depletion) should be rare amongst naturally occurring HIV-1 Vif variants. Furthermore, if effects on viral fitness are synergistic, such combinations should occur less frequently than predicted by chance. We therefore examined covariance of polymorphisms of residues 31 and

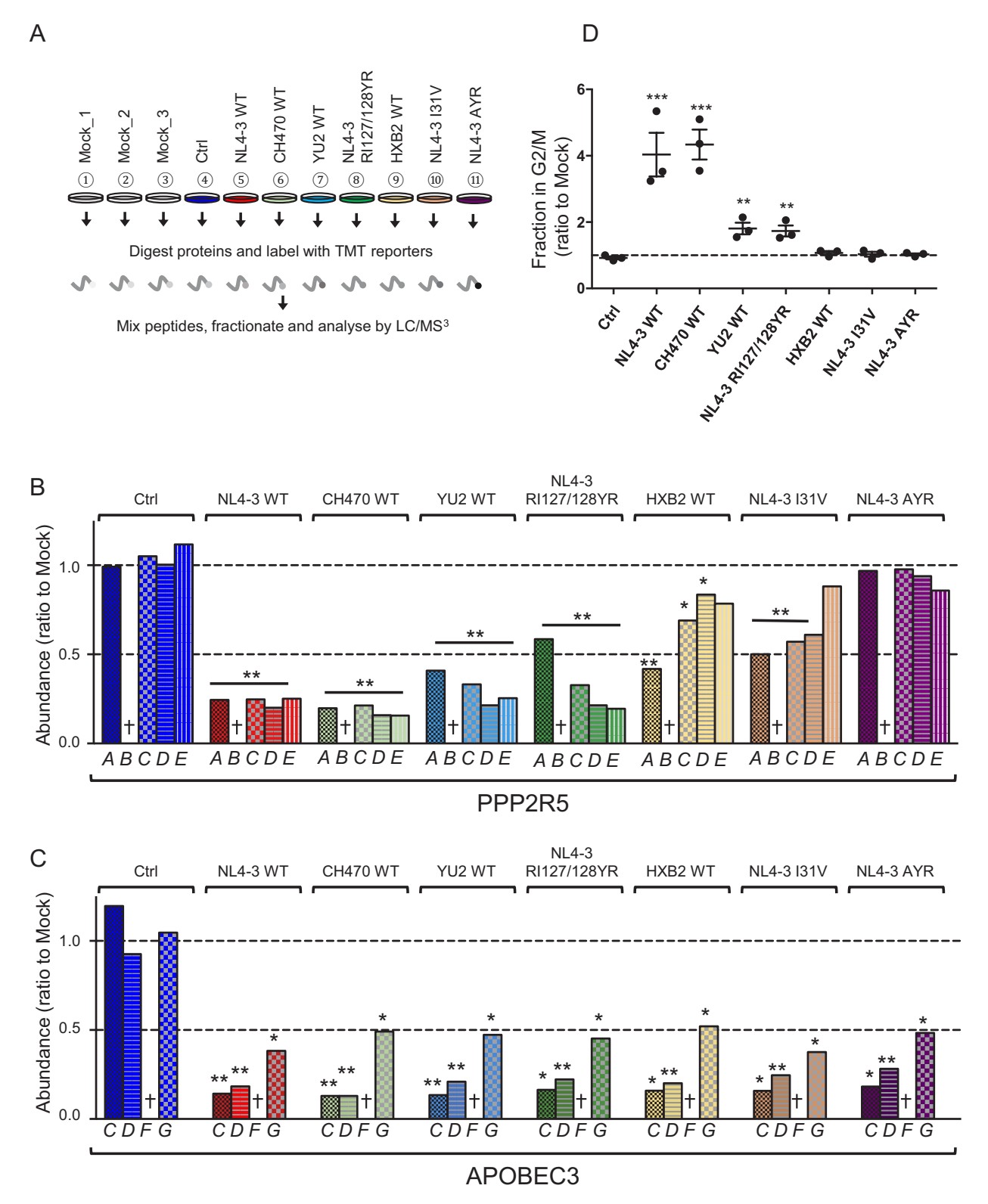

**Figure 6.** Depletion of endogenous APOBEC3 and PPP2R5 family proteins by naturally occurring HIV-1 Vif variants. (**A**) Overview of proteomic experiment 2 (naturally occurring Vif variants and corresponding point mutants). CEM-T4s were transduced with lentiviruses encoding EGFP-P2A-Vif at an MOI of 3, then analysed by TMT-based quantitative proteomics after 48 hr. Mock_1/2/3, biological replicates. Ctrl, control construct encoding EGFP. NL4-3 AYR, NL4-3 Vif with both I31A and RI127/128YR mutations. (**B–C**) Depletion of endogenous PPP2R5 family (**B**) or APOBEC3 family (**C**) proteins by

*Figure 6 continued on next page*

*Figure 6 continued*

naturally occurring Vif variants and corresponding point mutants in cells from (A). For each Vif variant or point mutant, abundance of respective PPP2R5 or APOBEC family members is shown as a ratio to the mean abundance of the same family member in the three mock-transduced samples. Significant outliers from the distribution of abundances in mock-transduced samples are highlighted (see Materials and methods and *Figure 2—figure supplement 2* for further details). *p<0.05; **p<0.005. † Not detected in this experiment (PPP2R5B, APOBEC3F). (D) Regulation of cell cycle by naturally occurring Vif variants and corresponding point mutants. CEM-T4s were transduced with lentiviruses encoding EGFP-P2A-Vif at an MOI of 3, then fixed in 90% methanol, stained with 7-AAD and analysed by flow cytometry after 48 hr. Individual data points reflect three biological replicates (representative data, *Figure 6—figure supplement 2*). **p<0.005. ***p<0.0005. Ctrl, control construct encoding EGFP. NL4-3 AYR, NL4-3 Vif with both I31A and RI127/128YR mutations. All other details as per *Figure 3C–D*.

The online version of this article includes the following source data and figure supplement(s) for figure 6:

**Source data 1.** Complete data from proteomic experiment 2 (naturally occurring Vif variants and corresponding point mutants).
**Figure supplement 1.** Sequence of Exonic Splicing Silencer of Vpr (ESSV) in NL4-3 and YU2 Vif variants.
**Figure supplement 2.** Regulation of cell cycle by naturally occurring Vif variants and corresponding point mutants (representative data).
**Figure supplement 3.** Depletion of PPP2R5B, APOBEC3G and APOBEC3F by Vif AYR.

128 across the clade B HIV-1 Vif sequences available from the Los Alamos National Laboratory Web Alignments database (*Figure 7D*; same 2171 sequences as *Figure 5A*).

Amongst these sequences, 21.8% encode 31V (less active) and 33.6% encode R128 (less active). By chance, combinations of 31V and 128R would therefore be expected in 7.3% of sequences. Conversely, this combination is observed in only 5.8% of sequences. Whist this difference appears modest, the association between these polymorphisms is highly statistically significant (p=0.0003, Fisher's exact test, *Figure 7—figure supplement 3A*, left panel). We observed similar, significant under-representation when we limited our analysis to clade B viruses encoding combinations of 31I/V and 128I/R (*Figure 7—figure supplement 3A*, right panel), or extended it to include Vif sequences from all (any clade, including B) non-recombinant HIV-1 M group viruses (*Figure 7—figure supplement 3B–C*; same 3412 sequences as *Figure 5—figure supplement 1*).

To distinguish functional covariance of these residues from background linkage disequilibrium (co-inheritance of polymorphisms from a common ancestor), we constructed phylogenetic trees of all Vif variants based on Vif (*Figure 7E*) or Nef, Gag or Env (*Figure 7—figure supplement 4A*). Regardless of the viral protein used, viruses encoding different combinations of 31/128 polymorphisms were scattered throughout the phylogeny, with no obvious founder effect. Again, similar results were seen when we extended our analysis to include Vif sequences from all (any clade, including B) non-recombinant HIV-1 M group viruses (*Figure 7—figure supplement 4B*).

Taken together, these data therefore provide evidence of a functional interaction between residues 31/128, and suggest significant in vivo selection pressure to maintain the ability of Vif to antagonise PP2A.

## Discussion

The study of cellular proteins and processes targeted by HIV has provided critical insights into the host-virus interaction. Typically, these targets have been identified piecemeal, using candidate approaches. In contrast, we have recently adopted unbiased proteomic approaches to identify novel substrates of HIV accessory proteins (*Greenwood et al., 2016*; *Greenwood et al., 2019*; *Matheson et al., 2015*; *Naamati et al., 2019*). A key challenge is now to determine the biological significance of these targets for HIV-infected cells: both *whether* they are important, and *why* they are important.

In this study, we sought to address these questions for Vif targets PPP2R5A-E. By demonstrating that depletion of PPP2R5 family subunits by Vif is separable from targeting of APOBEC3 family proteins, we formally prove that PP2A antagonism is neither required for, nor an epiphenomenon of, APOBEC3 family protein depletion. Combined with evidence of conservation across HIV-1 viruses and the broader lentiviral lineage (*Greenwood et al., 2016*), these observations provide strong genetic evidence for the importance of PPP2R5 depletion by Vif in vivo.

Strikingly, the critical residues for PPP2R5 depletion identified in our screen included several previously determined to be important for Vif-dependent cell cycle arrest in other, independent studies (31, 33, 44) (*DeHart et al., 2008*; *Zhao et al., 2015*). As well as residues required for CUL5 complex

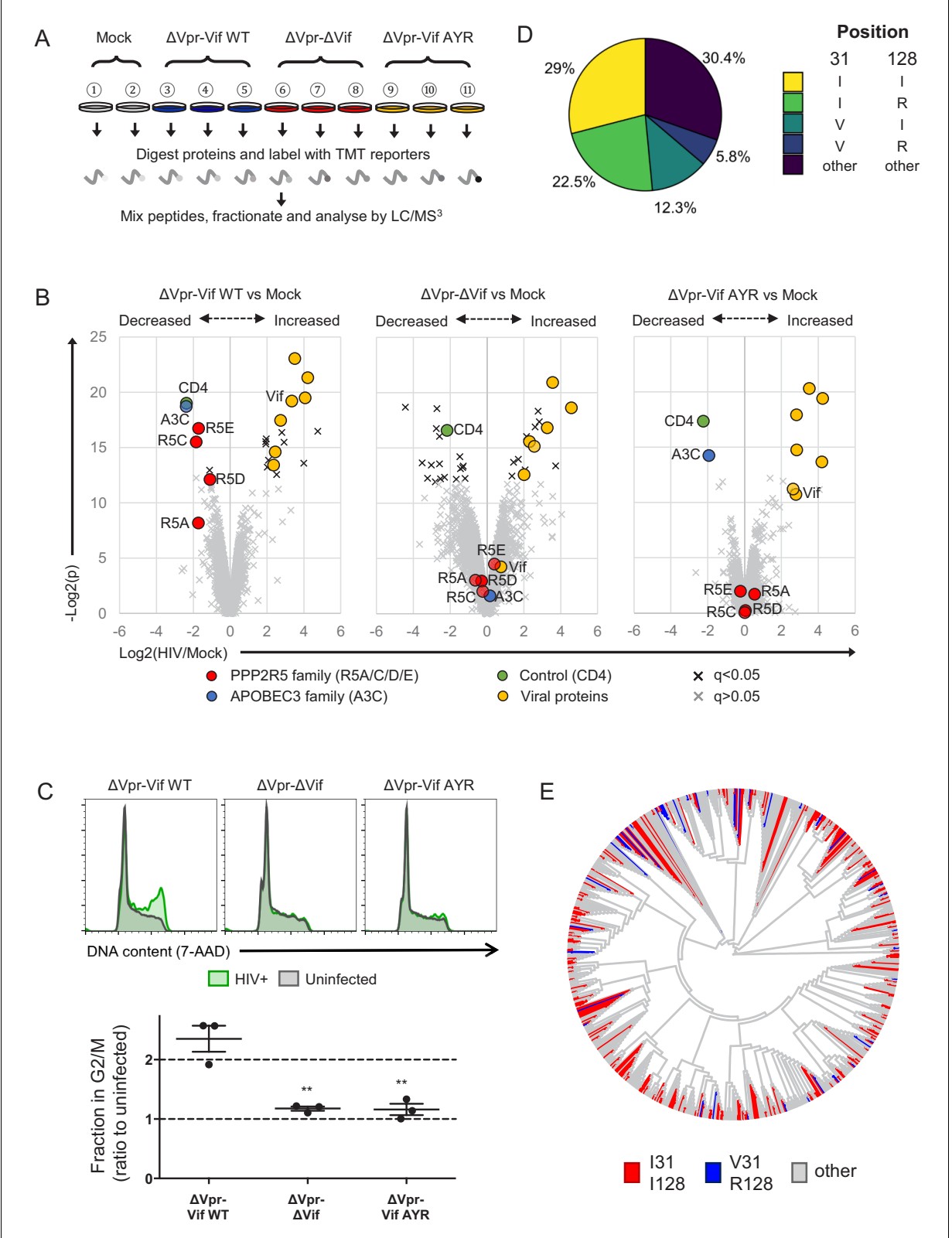

**Figure 7.** Selective regulation of PPP2R5 family subunits during HIV-1 infection. (**A**) Overview of proteomic experiment 3 (viral infections). CEM-T4s were infected with HIV-AFMACS viruses at an MOI of 0.5, then purified using AFMACS (*Figure 7—figure supplement 1A–B*) and analysed by TMT-based quantitative proteomics after 48 hr. Biological replicates are shown. Vif AYR, NL4-3 Vif with both I31A and RI127/128YR mutations. (**B**) Protein abundances in HIV-infected vs mock-infected cells from (**A**). Volcano plots show statistical significance (*y*-axis) vs fold change (*x*-axis) for 6294 viral and

*Figure 7 continued on next page*

*Figure 7 continued*

cellular proteins (no missing values). Pair-wise comparisons of mock-infected cells with cells infected with ΔVpr-Vif WT (left panel), ΔVpr-ΔVif (middle panel) or ΔVpr-Vif AYR (right panel) viruses are shown. Proteins with Benjamini-Hochberg FDR-adjusted p values (q values)<0.05 (black crosses) or >0.05 (grey crosses) are indicated (FDR threshold of 5%). Proteins highlighted in each plot are summarised in the key. 4 out of 5 PPP2R5 family subunits (A/C/D/E) were quantitated, but only 1 out of 7 APOBEC3 family members (C). (C) Regulation of cell cycle in HIV-infected CEM-T4s. Cells were infected with HIV-AFMACS viruses at an MOI of 0.5, then stained with FITC-conjugated anti-LNGFR antibody, fixed in 90% methanol, stained with 7-AAD and analysed by flow cytometry after 48 hr. Representative data (upper panels) from three biological replicates (lower panel) are shown. Green, LNGFR+ cells (HIV+); grey, LNGFR- cells (uninfected). For each virus, the fraction of HIV+ (LNGFR+) cells in G2/M is shown as a ratio to the fraction of uninfected (LNGFR-) cells in G2/M. Individual data points reflect biological replicates. Mean values with SEM are indicated. Significant differences are highlighted for each pair-wise comparison (t-tests). **$p<0.005$. (D) Pair-wise combinations of key amino acid polymorphisms amongst 2171 naturally occurring HIV-1 M group Vif variants (clade B). Frequencies of amino acids corresponding to residues 31 and 128 of NL4-3 Vif are shown. An equivalent pie chart for all naturally occurring non-recombinant HIV-1 M group Vif variants (all clades) is shown in *Figure 7—figure supplement 3B*. (E) Phylogenetic tree of 795 HIV-1 M group viruses (clade B) with protein sequences available for all of Vif, Gag, Env and Nef (based on relatedness of Vif). Viruses encoding Vif variants with I31/I128 (most active, red) and V31/R128 (least active, blue) are highlighted. Equivalent phylogenetic trees based on relatedness of Gag, Env or Nef are shown in *Figure 7—figure supplement 4A*.

The online version of this article includes the following source data and figure supplement(s) for figure 7:

**Source data 1.** Complete data from proteomic experiment 3 (viral infections).
**Figure supplement 1.** AFMACS-based purification of infected cells for proteomic experiment 3 (viral infections).
**Figure supplement 2.** Regulation of cell cycle in HIV-infected primary human CD4+ T cells.
**Figure supplement 3.** Additional bioinformatics analysis.
**Figure supplement 4.** Additional phylogenetic trees.

assembly (114 and 145), several additional residues (14, 36, 48 and 40) were implicated in the same studies. Amongst these, a K36A point mutant showed an intermediate effect on PPP2R5B depletion in our screen (*Figure 1—figure supplement 2A*). The other residues were not tested, because we focussed on regions of Vif not known to be important for depletion of APOBEC3 family proteins, and residues with solvent-exposed side chains unlikely to lead to structural disruption.

We were initially puzzled because some Vif point mutants were markedly impaired in their ability to cause cell cycle arrest, yet retained the ability to deplete at least some PPP2R5 family subunits. Furthermore, the ability of Vif to cause cell cycle arrest did not appear to correlate with depletion of any one, specific PPP2R5 subunit. In fact, because efficient depletion of all PPP2R5 subunits is required to halt cell cycle progression, these are not paradoxes at all. This same model also suggests explanations for two related phenomena.

First, expression of HIV-1 Vif in mouse or COS cells results in depletion of PPP2R5D, but does not cause cell cycle arrest (*Evans et al., 2018*). As with Vif point mutants and naturally occurring variants in human cells, it seems likely that another PPP2R5 subunit escapes depletion in these species-mismatched cells. Second, we previously found the ability of Vif to antagonise at least some PPP2R5 subunits to be widely conserved (*Greenwood et al., 2016*), but the ability to cause cell cycle arrest is variable amongst HIV-1 Vif variants (*Evans et al., 2018*; *Zhao et al., 2015*). Since efficient depletion of all expressed PPP2R5 subunits is required to cause cell cycle arrest, escape of even a single subunit allows cell cycle to progress.

Amongst all HIV-1 Vif sequences analysed here, the commonest single combination of residues at positions 31 and 128 was 31I/128I, accounting for approximately 30% of Vif variants. Most of these sequences also encode 33R or 33K and, like NL4-3 and CH470 Vif, are therefore expected to efficiently degrade PPP2R5A-E, and cause cell cycle arrest. Conversely, only approximately 5% encode 31V/128R, and are therefore expected to be severely impaired for PPP2R5A-E degradation. It is likely that most of the remaining Vif variants, like YU2 and HXB2, are active against at least some PPP2R5 family members, but may be variably attenuated in their ability to cause cell cycle arrest. Interestingly, naturally occurring Vif variants have also been shown to exhibit a spectrum of activity against APOBEC3 family proteins, including variants which fail to neutralise one or more APOBEC3 family proteins (*Binka et al., 2012*; *Iwabu et al., 2010*; *Mulder et al., 2008*; *Simon et al., 2005*).

Mechanistically, hyperphosphorylation of aurora kinase substrates is expected to contribute significantly to Vif-dependent cell cycle arrest (*Foley et al., 2011*). Nonetheless, PP2A is a 'master regulator' of cell cycle (*Wlodarchak and Xing, 2016*), and depletion of PPP2R5A-E by Vif causes widespread remodelling of the phosphoproteome, implying activation of multiple kinases (*Greenwood et al., 2016*). Consistent with this, Vif-dependent cell cycle arrest was previously shown

to require TP53 (*Izumi et al., 2010*), and several studies have identified upstream regulation of TP53 by PP2A in different systems (*Ajay et al., 2010*; *Li et al., 2002*; *Yang and Phiel, 2010*).

Many RNA and DNA viruses cause and are thought to benefit from cell cycle arrest (*Bagga and Bouchard, 2014*). In HIV infection, G2/M cell cycle arrest was first attributed to Vpr (*He et al., 1995*; *Jowett et al., 1995*; *Rogel et al., 1995*), which remains better known for this function. Early studies suggested a positive effect of G2/M arrest on transcription from the HIV-1 LTR (*Goh et al., 1998*; *Gummuluru and Emerman, 1999*), but more recent results have raised the possibility that cell cycle arrest may be secondary to another Vpr-dependent process, such as antagonism of innate immunity (*Laguette et al., 2014*). Nonetheless, targeting of the same cell biological process by multiple viral accessory proteins is strong a priori evidence of biological importance in vivo.

Functional redundancy with Vpr may also help explain why Vif-dependent cell cycle arrest is not more strictly conserved across naturally occurring HIV-1 Vif variants. In addition, key polymorphic residues which regulate PPP2R5 antagonism may be subject to balancing selection pressures. First, the requirement to maintain the Exonic Splicing Silencer of Vpr (ESSV) at the RNA level limits the sequence variability tolerated at position 128 (*Madsen and Stoltzfus, 2005*). Second, polymorphisms at position 31 also regulate antagonism of APOBEC3H (*Zhao et al., 2015*). Indeed, at least in some cases, the abilities of Vif to antagonise APOBEC3H and cause cell cycle arrest were found to be mutually exclusive.

Vif-dependent cell cycle arrest correlates with viral cytopathicity (*Evans et al., 2018*; *Sakai et al., 2006*), and was reported to enhance HIV-1 replication in vitro in a previous study using chimeric HXB2/NL4-3 Vif variants (*Izumi et al., 2010*). Classic experiments comparing WT and Vif-deficient viruses in permissive cells often examined HXB2 (*Gabuzda et al., 1992*) or YU2 (*Gaddis et al., 2003*) viruses. Since these Vif variants are shown here to be attenuated in their activity against different PPP2R5 subunits, manifested by a reduced ability to cause cell cycle arrest, it is likely that these studies failed to fully capture the effects of PP2A antagonism on viral infection.

As importantly, the ability to assess APOBEC3-independent effects of Vif on viral fitness in vitro has hitherto been limited to comparisons between WT and Vif-deficient viruses on an APOBEC3-negative background, such as the CEM-SS cell line. In contrast, the point mutants identified in this study maintain the ability to antagonise APOBEC3 family proteins, and will therefore allow the assessment of Vif-dependent PP2A antagonism by the community in a full range of cell types, including primary and myeloid cells, as well as providing a mechanistic framework to interpret the results.

# Materials and methods

**Key resources table**

| Reagent type (species) or resource | Designation | Source or reference | Identifiers | Additional information |
|---|---|---|---|---|
| Cell line (human) | CEM-T4 T cells (CEM-T4s) | NIH AIDS Reagent Program | Cat. #: 117 | Also known as CEM CD4+ cells |
| Cell line (human) | THP-1 cells (THP-1s) | NIH AIDS Reagent Program | Cat. #: 9942 | Used for cDNA library generation |
| Cell line (human) | HeLa cells (HeLas) | Lehner laboratory stocks | RRID:CVCL_0030 | |
| Cell line (human) | HEK 293T cells (293Ts) | Lehner laboratory stocks | RRID:CVCL_0063 | |
| Antibody | Mouse monoclonal BV421-conjugated anti-CD4 | BioLegend | Cat. #: 317434 | Flow cytometry (1:50) |
| Antibody | Mouse monoclonal PE-conjugated anti-CD4 | BD Biosciences | Cat. #: 561843 | Flow cytometry (1:50) |
| Antibody | Mouse monoclonal AF647-conjugated anti-LNGFR | BioLegend | Cat. #: 345114 | Flow cytometry (1:50) |
| Antibody | Mouse monoclonal FITC-conjugated anti-LNGFR | BioLegend | Cat. #: 345103 | Flow cytometry (1:50) |
| Antibody | Mouse monoclonal BV421-conjugated anti CD4 | BioLegend | Cat. #: 317434 | Flow cytometry (1:50) |

*Continued on next page*

*Continued*

| Reagent type (species) or resource | Designation | Source or reference | Identifiers | Additional information |
|---|---|---|---|---|
| Antibody | Mouse monoclonal DyLight 650-conjugated anti-HA tag | Abcam | Cat. #: ab117515 | Flow cytometry (1:400) |
| Antibody | Rabbit monoclonal anti-PPP2R5D | Abcam | Cat. #: ab188323 | Immunoblot (1:5000) |
| Antibody | Mouse monoclonal anti-HIV-1 Vif | NIH AIDS Reagent Program | Cat. #: 6459 | Immunoblot (1:2500) |
| Antibody | Rabbit polyclonal anti-FMR1 (FMRP) | Cell Signalling Technology | Cat. #: 4317 | Immunoblot (1:1000) |
| Antibody | Rabbit polyclonal anti-DPH7 | Atlas Antibodies | Cat. #: HPA022911 | Immunoblot (1:1000) |
| Antibody | Mouse monoclonal anti-β-actin | Sigma | Cat. #: A5316 | Immunoblot (1:20000) |
| Antibody | Mouse monoclonal anti-p97 (VCP) | Abcam | Cat. # ab11433 | Immunoblot (1:10000) |
| Antibody | Rabbit polyclonal anti-total AURKB | Cell Signalling Technology | Cat. #: 3094 | Immunoblot (1:500) |
| Antibody | Rabbit monoclonal anti-phospho-AURK | Cell Signalling Technology | Cat. #: 2914 | Immunoblot (1:500) |
| Antibody | Rabbit polyclonal anti-GFP | Thermo Scientific | Cat. #: A-11122 | Immunoblot (1:2500) |
| Recombinant DNA reagent | pHRSIN-SE-P2A-SBP-ΔLNGFR-W | *Matheson et al., 2014* | N/A | Used as a control and to express codon optimised Vif variants |
| Recombinant DNA reagent | pHRSIN-SE-W-pSV40-puro | *van den Boomen et al., 2014* | N/A | Used as a control |
| Recombinant DNA reagent | pHRSIN-S-W-pGK-puro | *Greenwood et al., 2016* | N/A | Used to express HA-tagged PPP2R5B, APOBEC3F and APOBEC3G |
| Recombinant DNA reagent | HIV-AFMACS | *Naamati et al., 2019* | GenBank: MK435310.1 | pNL4-3-ΔEnv-Nef-P2A-SBP-ΔLNGFR proviral construct |
| Recombinant DNA reagent | V245 pCEP-4HA B56alpha | Addgene | Cat. #: 14532 | Standard for quantification of PPP2R5A mRNA |
| Recombinant DNA reagent | V245 pCEP-4HA B56beta | Addgene | Cat. #: 14533 | Standard for quantification of PPP2R5B mRNA |
| Recombinant DNA reagent | V245 pCEP-4HA B56gamma1 | Addgene | Cat. #: 14534 | Standard for quantification of PPP2R5C mRNA |
| Recombinant DNA reagent | V245 pCEP-4HA B56delta | Addgene | Cat. #: 14536 | Standard for quantification of PPP2R5D mRNA |
| Recombinant DNA reagent | V245 pCEP-4HA B56epsilon | Addgene | Cat. #: 14537 | Standard for quantification of PPP2R5E mRNA |
| Recombinant DNA reagent | TBP cDNA clone: IRATp970C11110D | Source Bioscience | GenBank: BC110341.1 | Standard for quantification of TBP mRNA |
| Commercial assay or kit | NEBuilder HiFi DNA Assembly Cloning Kit | NEB | Cat. #: E5520S | |
| Commercial assay or kit | Fugene 6 Transfection Reagent | Promega | Cat. #E2691 | |
| Commercial assay or kit | Lipofectamine RNAiMAX Transfection Reagent | Invitrogen | Cat. #: 18080044 | |
| Chemical compound, drug | Lenti-X Concentrator | Clontech | Cat. #: 631232 | |
| Commercial assay or kit | Dynabeads Biotin Binder | Invitrogen | Cat. #: 11047 | |

*Continued on next page*

*Continued*

| Reagent type (species) or resource | Designation | Source or reference | Identifiers | Additional information |
|---|---|---|---|---|
| Commercial assay or kit | Dynabeads Untouched Human CD4 T Cells kit | Invitrogen | Cat. #: 11346D | |
| Commercial assay or kit | Dynabeads Human T-Activator CD3/CD28 | Invitrogen | Cat. #: 11132D | |
| Commercial assay or kit | S-Trap micro MS Sample Preparation Kit | Protifi | Cat. #: C02-micro | |
| Commercial assay or kit | TMT10plex Isobaric Label Reagent Set | Thermo Scientific | Cat. #: 90110 | |
| Commercial assay or kit | Superscript III First-Strand Synthesis System | Invitrogen | Cat. #: 18080051 | |
| Software, algorithm | PyMOL Molecular Graphics System, Version 2.0 | Schrödinger | RRID:SCR_006054 | https://www.schrodinger.com/pymol |
| Software, algorithm | Proteome Discoverer 2.1 | Thermo Scientific | RRID:SCR_014477 | |
| Software, algorithm | R v.3.5.3 | *R Development Core Team, 2019* | RRID:SCR_001905 | https://www.R-project.org/ |
| Software, algorithm | limma | *Ritchie et al., 2015* | RRID:SCR_010943 | https://bioconductor.org/packages/limma/ |
| Software, algorithm | WebLogo | *Crooks et al., 2004* | RRID:SCR_010236 | http://weblogo.berkeley.edu |
| Software, algorithm | seqinr | *Charif and Lobry, 2007* | N/A | https://cran.r-project.org/web/packages/seqinr/ |
| Software, algorithm | ggplot2 | *Wickham, 2009* | RRID:SCR_014601 | https://ggplot2.tidyverse.org |
| Software, algorithm | ggtree | *Yu et al., 2018* | N/A | https://bioconductor.org/packages/release/bioc/html/ggtree.html |
| Software, algorithm | Clustal Omega | *Sievers and Higgins, 2014* | RRID:SCR_001591 | https://www.ebi.ac.uk/Tools/msa/clustalo/ |
| Software, algorithm | Prism 7.0 | GraphPad | RRID:SCR_002798 | |

## General cell culture

CEM-T4 T cells (*Foley et al., 1965*) and THP-1 cells (*Wu et al., 2004*) were obtained directly (<1 year) from the AIDS Reagent Program, Division of AIDS, NIAD, NIH (Dr J. P. Jacobs, and Drs L. Wu and V. N. KewalRamani, respectively), and cultured at a density of $5 \times 10^5$ to $2 \times 10^6$ cells/ml in RPMI supplemented with 10% fetal calf serum (FCS), 100 units/ml penicillin and 0.1 mg/ml streptomycin at 37°C in 5% CO2. HeLa cells and HEK 293T cells (authenticated by STR profiling [*Menzies et al., 2018*; *Miles et al., 2017*]) were obtained from Lehner laboratory stocks and cultured in DMEM supplemented with 10% FCS, 100 units/ml penicillin and 0.1 mg/ml streptomycin at 37°C in 5% CO2. All cells were regularly screened and confirmed to be mycoplasma negative (Lonza MycoAlert).

## Primary cell isolation and culture

Primary human CD4+ T cells were isolated from peripheral blood by density gradient centrifugation over Lympholyte-H (Cedarlane Laboratories) and negative selection using the Dynabeads Untouched Human CD4 T Cells kit (Invitrogen) according to the manufacturer's instructions. Purity was assessed by flow cytometry for CD3 and CD4 and routinely found to be ≥95%. Cells were activated using Dynabeads Human T-Activator CD3/CD28 beads (Invitrogen) according to the manufacturer's instructions and cultured in RPMI supplemented with 10% FCS, 30 U/ml recombinant human IL-2 (PeproTech), 100 units/ml penicillin and 0.1 mg/ml streptomycin at 37°C in 5% CO2.

## Ethics statement

Ethical permission for this study was granted by the University of Cambridge Human Biology Research Ethics Committee (HBREC.2017.20). Written informed consent was obtained from all volunteers prior to providing blood samples.

## Vectors for transgene expression

Sequences for Vif variants from NL4-3 (AF324493.2), HXB2 (K03455.1), YU2 (GenBank: M93258.1) and CH470 (JX972238-JX972249) viruses were obtained from GenBank. The CH470 transmitted founder (TF) Vif sequence was inferred as previously described (*Fenton-May et al., 2013*; *Liu et al., 2013*; *Parrish et al., 2013*).

For co-expression of codon optimised Vif variants with EGFP, gBlocks (IDT) encoding NL4-3, HXB2, YU2 or CH470 Vif were incorporated into pHRSIN-SE-P2A-SBP-ΔLNGFR-W (*Matheson et al., 2014*) in place of SBP-ΔLNGFR by Gibson assembly between XhoI/KpnI sites (generating pHRSIN-SE-P2A-Vif-W vectors). In these vectors, Vif variants are expressed from the *Friend spleen focus-forming virus* (SFFV) promoter as EGFP-P2A-Vif, downstream of EGFP and a 'self-cleaving' *Porcine teschovirus-1 2A* (P2A) peptide.

Complete sequences for all gBlocks are included in *Supplementary file 1* (Codon-optimised Vif variants synthesised as gBlocks). Codon optimisation was conducted using the IDT codon optimisation tool, and sequences were verified by Sanger sequencing (Source BioScience).

The parental vector (in which EGFP and the SBP-ΔLNGFR cell surface selection marker are expressed from the SFFV promoter as EGFP-SBP-ΔLNGFR) was used here as a control. Where indicated, pHRSIN-SE-W-pSV40-puro (in which EGFP is expressed from the SFFV promoter as a single transgene) was used as an alternative control (*van den Boomen et al., 2014*).

To generate stable 293T cell lines for our flow cytometric screen, N-terminal 4xHA-tagged PPP2R5B and C-terminal 4xHA-tagged APOBEC3G were expressed using pHRSIN-S-W-pGK-puro exactly as previously described (*Greenwood et al., 2016*). HeLas stably expressing exogenous PPP2R5B were generated using the same construct. C-terminal 4xHA-tagged APOBEC3F was amplified by PCR from a cDNA library generated from THP-1s using the Superscript III First-Strand Synthesis System (Invitrogen) with Oligo(dT) (Invitrogen), and also expressed using pHRSIN-S-W-pGK puro. The complete sequence is included in *Supplementary file 1* (C-terminal 4xHA-tagged APOBEC3F coding sequence in pHRSIN-S-W-pGK puro), and was verified by Sanger sequencing (Source BioScience).

## Vif mutant library construction

To generate a library of Vif point mutants, a PCR and Gibson assembly-based approach was used to modify codon-optimised NL4-3 Vif directly in pHRSIN-SE-P2A-Vif-W (*Figure 1—figure supplement 1A*).

Briefly, forward and reverse primers encoding each point mutation were designed with ~15 bp fully complementary flanking sequences. These mutation-specific primers were used in pairwise PCR reactions in conjunction with common primers complementary to the vector backbone, which was cut between XhoI/KpnI sites. The two PCR products were then assembled into the vector using the NEBuilder HiFi DNA Assembly Master Mix (NEB).

Sequences for all primers used are tabulated in *Supplementary file 1* (PCR primers for Vif mutant library construction). All sequences were verified by Sanger sequencing (Source BioScience).

## HIV-1 molecular clones

HIV-AFMACS (pNL4-3-ΔEnv-Nef-P2A-SBP-ΔLNGFR; GenBank: MK435310.1) has been previously described (*Naamati et al., 2019*). To introduce mutations in the native NL4-3 Vif coding sequence, the same PCR and Gibson assembly-based approach developed for Vif mutant library construction was used, cutting the vector backbone between AgeI/SalI sites. Where indicated, multiple mutations were introduced sequentially.

To generate ΔVpr-Vif WT virus (lacking Vpr expression, but encoding WT Vif), a silent mutation was introduced into Vif codon 173 (AGA >AGG; both encoding Arg), eliminating the Vpr start codon in the +two reading frame. Additional point mutations were introduced to generate ΔVpr-Vif AYR virus (lacking Vpr expression, but encoding Vif with I31A and R127Y/I128R mutations) and ΔVpr-ΔVif

virus (lacking Vpr expression, and encoding two premature stop codons after the final in-frame start codon in the Vif open reading frame).

Final Vif coding sequences for each virus are included in *Supplementary file 1* (Vif coding sequences in HIV-AFMACS viruses). Sequences were verified by Sanger sequencing (Source BioScience).

## Transient transfection

For the flow cytometric screen, 293T cells stably expressing HA-tagged PPP2R5B or APOBEC3G were transfected with 200 ng/well control or Vif expression vector in 24-well plates using FuGENE 6 (Promega). After 36 hr, cells were harvested with trypsin-EDTA and analysed by flow cytometry.

## siRNA transfection

For RNAi-mediated knockdown, HeLa cells were transfected with custom siRNA duplexes (Sigma) using transfected using Lipofectamine RNAiMAX (Invitrogen) according to the manufacturer's instructions.

Briefly, $2 \times 10^5$ cells/well were seeded in 6-well plates 24 hr prior to transfection with a total of 50 pmol/well siRNA (individual or pooled). Knockdown was verified by real-time PCR or immunoblot 24 hr post-transfection, and cells were re-seeded prior to cell cycle analysis 48 hr post-transfection (target 50% confluency).

All siRNA target sequences used are tabulated in *Supplementary file 1* (Target sequences for RNAi). Cells not subjected to knockdown were transfected with MISSION siRNA Universal Negative Control #1 (Sigma) at equivalent concentrations.

## Viral stocks

VSVg-pseudotyped lentivector stocks were generated by co-transfection of 293Ts with pHRSIN-based lentivector, p8.91 and pMD.G at a ratio of 2:1:1 (µg) DNA and a DNA:FuGENE 6 ratio of 1 µg:3 µl. Media was changed the next day and viral supernatants harvested and filtered (0.45 µm) at 48 hr prior to concentration with Lenti-X Concentrator (Clontech) and storage at −80˚C.

VSVg-pseudotyped HIV-AFMACS viral stocks were generated by co-transfection of 293Ts with HIV-AFMACS molecular clones and pMD.G at a ratio of 9:1 (µg) DNA and a DNA:FuGENE 6 ratio of 1 µg:6 µl. Viral supernatants were harvested, filtered, concentrated and stored as per pHRSIN-based lentivector stocks.

Lentivector/viral stocks were titrated by transduction/infection of known numbers of relevant target cells with known volumes of stocks under standard experimental conditions, followed by flow cytometry for EGFP (GFP-expressing lentivectors) or SBP-ΔLNGFR and CD4 (HIV-AFMACS viruses) at 48 hr to identify the fraction of transduced/infected cells (f) containing at least one transcriptionally active provirus (EGFP positive or SBP-ΔLNGFR positive/CD4 low). The number of transducing/ infectious units present was then calculated by assuming a Poisson distribution (where $f = 1-e^{-MOI}$). Typically, a dilution series of each stock was tested, and titre determined by linear regression of -ln (1-f) on volume of stock.

## Transductions and infections

Primary human CD4+ T cells, CEM-T4s, HeLas and 293Ts were transduced or infected by spinoculation at 800 g for 1 hr in a non-refrigerated benchtop centrifuge in complete media supplemented with 10 mM HEPES. Stable cell lines were selected using puromycin, and 293Ts expressing HA-tagged APOBEC3F or APOBEC3G were single-cell cloned prior to flow cytometric screening.

## Antibody-Free magnetic cell sorting (AFMACS)

AFMACS-based selection of CEM-T4s using the streptavidin-binding SBP-ΔLNGFR affinity tag was carried out essentially as previously described (*Matheson et al., 2014*; *Naamati et al., 2019*). Briefly, $1 \times 10^6$ CEM-T4s/condition were infected with VSV-g pseudotyped HIV-AFMACS viruses at an MOI of 0.5. 48 hr post-infection, washed cells were resuspended in incubation buffer (IB; Hank's balanced salt solution, 2% dialysed FCS, 1x RPMI Amino Acids Solution (Sigma), 2 mM L-glutamine, 2 mM EDTA and 10 mM HEPES) at $10^7$ cells/ml and incubated with Dynabeads Biotin Binder (Invitrogen) at a bead-to-total cell ratio of 4:1 for 30 min at 4˚C. Bead-bound cells expressing SBP-ΔLNGFR

were selected using a DynaMag-2 magnet (Invitrogen), washed to remove uninfected cells, then released from the beads by incubation in complete RPMI with 2 mM biotin for 15 min at room temperature (RT). Enrichment was assessed by flow cytometry pre- and post-selection.

## Proteomics

### Sample preparation

For TMT-based whole cell proteomic analysis of transduced or infected CEM-T4s, washed cell pellets were lysed in 50 mM HEPES pH 8 with 5% SDS followed by 10 min (30 s on/off) sonication in a Bioruptor Pico sonicator (Diagenode) at 18˚C. Lysates were quantified by BCA assay (Thermo Scientific) and 25 µg (transduced CEM-T4s, experiments 1–2) or 10 µg (infected and AFMACS-selected CEM-T4s, experiment 3) total protein/condition used for further analysis.

Sample volumes were equalised with lysis buffer and proteins reduced and alkylated by addition of 10 mM TCEP and 20 mM iodoacetamide followed by incubation at RT for 30 min, protected from light. Samples were then processed using S-Trap micro columns (Protifi). To each sample 10% v/v $H_3PO_4$ was added and samples mixed by vortexing briefly. 6 volumes of 90% methanol HEPES pH 7.1 (loading buffer) were then added and pipette-mixed before loading onto columns using a vacuum manifold.

Samples were then washed with 4 × 150 µl loading buffer. A 1:25 enzyme:protein ratio of LysC/trypsin mix (Promega) was added to each column in 30 µl of 50 mM HEPES pH 8 with 0.1% sodium deoxycholate (SDC). Columns were placed into microcentrifuge tubes and incubated for 6 hr at 37˚C in a Thermomixer S (Eppendorf) without shaking. Open tubes of water were placed in empty positions and the Thermomixer lid used to minimise evaporation.

After incubation, peptides were eluted in three stages: 40 µl 10 mM HEPES pH 8; 35 µl 0.2% formic acid (FA); then 35 µl 0.2% FA in 50% Acetonitrile (ACN). Samples were dried for a short period in a vacuum centrifuge to evaporate ACN and then acidified with FA to precipitate SDC. Samples were then made up to ~100 µl with water, then 600 µl ethyl acetate was added and samples vortexed vigorously. After centrifugation at 15000 g for 5 min the lower phase (containing peptides) was retained and the upper phase (containing SDC and ethyl acetate) was discarded.

After drying fully in a vacuum centrifuge, samples were resuspended in 21 µl 100 mM HEPES pH 8, to which was added 0.2 mg of TMT label dissolved in 9 µl ACN. After 1 hr incubation at RT samples were analysed by LCMS to ensure complete labelling, then pooled and dried by ~50% in a vacuum centrifuge. The pooled sample was made up to ~1 ml in a final concentration of 0.1% triflouracetic acid (TFA) and the pH was adjusted to <2 with FA. The samples were then subjected to C18 SPE clean-up using 500 mg Sep-Pak tC18 cartridges (Waters). Columns were wetted with 1 ml ACN and equilibrated with 3 ml 0.1% TFA before loading the sample, washing with 2 ml 0.1% TFA and eluting with 250 µl 40% ACN, 250 µl 80% ACN and 250 µl 80% ACN. The eluates were dried in a vacuum centrifuge.

### Off-line high pH reversed-phase (HpRP) peptide fractionation

HpRP fractionation was conducted on an Ultimate 3000 UHPLC system (Thermo Scientific) equipped with a 2.1 mm ×15 cm, 1.7 µm Kinetex EVO C18 column (Phenomenex). Solvent A was 3% ACN, Solvent B was 100% ACN, and solvent C was 200 mM ammonium formate (pH 10). Throughout the analysis solvent C was kept at a constant 10%. The flow rate was 400 µl/min and UV was monitored at 280 nm. Samples were loaded in 90% A for 10 min before a gradient elution of 0–10% B over 10 min (curve 3), 10–34% B over 21 min (curve 5), 34–50% B over 5 min (curve 5) followed by a 10 min wash with 90% B. 15 s (100 µl) fractions were collected throughout the run. Peptide-containing fractions were orthogonally recombined into 24 (transduced CEM-T4s, experiments 1–2) or 12 (infected and AFMACS-selected CEM-T4s, experiment 3) fractions, dried in a vacuum centrifuge and stored at −20˚C prior to analysis.

### Mass spectrometry

Data were acquired on an Orbitrap Fusion mass spectrometer (Thermo Scientific) coupled to an Ultimate 3000 RSLC nano UHPLC (Thermo Scientific). Solvent A was 0.1% FA and solvent B was ACN/0.1% FA. HpRP fractions were resuspended in 20 µl 5% DMSO 0.5% TFA and 10 µl injected. Fractions were loaded at 10 µl/min for 5 min on to an Acclaim PepMap C18 cartridge trap column (300

μm × 5 mm, 5 μm particle size) in 0.1% TFA. After loading, a linear gradient of 3–32% B over 3 hr was used for sample separation over a column of the same stationary phase (75 μm × 50 cm, 2 μm particle size) before washing with 90% B and re-equilibration. An SPS/MS3 acquisition was used for all samples and was run as follows. MS1: quadrupole isolation, 120000 resolution, $5 \times 10^5$ AGC target, 50 msec maximum injection time, ions injected for all parallelisable time. MS2: quadrupole isolation at an isolation width of m/z 0.7, CID fragmentation (NCE 35) with the ion trap scanning out in rapid mode from m/z 120, $8 \times 10^3$ AGC target, 70 msec maximum injection time, ions accumulated for all parallelisable time. In synchronous precursor selection mode the top 10 MS2 ions were selected for HCD fragmentation (65NCE) and scanned out in the orbitrap at 50000 resolution with an AGC target of $2 \times 10^4$ and a maximum accumulation time of 120 msec, ions were not accumulated for all parallelisable time. The entire MS/MS/MS cycle had a target time of 3 s. Dynamic exclusion was set to + /− 10 ppm for 90 s, MS2 fragmentation was trigged at $5 \times 10^3$ ions.

### Data processing

Spectra were searched using Mascot within Proteome Discoverer 2.2 in two rounds. The first search was against the UniProt human reference proteome, a custom HIV proteome (adjusted to include the exact protein coding sequences used) and a compendium of common contaminants (Global Proteome Machine). The second search took all unmatched spectra from the first search and searched against the human trEMBL database. The following search parameters were used. MS1 tol: 10 ppm; MS2 tol: 0.6 Da; fixed mods: carbamidomethyl (C) and TMT (N-term, K); var mods: oxidation (M); enzyme: trypsin (/P). MS3 spectra were used for reporter ion based quantitation with a most confident centroid tolerance of 20 ppm. PSM FDR was calculated using Mascot percolator and was controlled at 0.01% for 'high' confidence PSMs and 0.05% for 'medium' confidence PSMs. Normalisation was automated and based on total s/n in each channel.

The mass spectrometry proteomic data have been deposited to the ProteomeXchange consortium via the PRIDE (*Perez-Riverol et al., 2019*), partner repository with the dataset identifier PXD018271 and are summarised in *Figure 2—source data 1*, *Figure 6—source data 1* and *Figure 7—source data 1*.

### Statistical analysis

Abundances or proteins/peptides satisfying at least a 'medium' FDR confidence were subjected to further analysis in Excel 2016 (Microsoft) and R v.3.6.1 (*R Development Core Team, 2019*).

For proteomic experiments 1 and 2, abundances in the 3 mock-transduced samples were used to calculate sample means ($\bar{x}$) and standard deviations (S) for each protein. Corresponding protein abundances in transduced cells were then compared with these values to determine standard scores (*t*-scores) for each condition: $(X-\bar{x})/S$ (where X represents protein abundance in the condition of interest). Significant outliers were identified by calculating two-tailed *p*-values using a *t*-distribution with 2 degrees of freedom. Illustrative *t*-score/*p*-value calculations for PPP2R5A in cells transduced with WT Vif or a control lentivector are shown in *Figure 2—figure supplement 2*.

For proteomic experiment 3, mean protein abundances in cells infected with ΔVpr-Vif WT, ΔVpr-Δ Vif, or ΔVpr-Vif AYR viruses were compared with mean protein abundances in mock-infected cells. For each pair-wise comparison, a moderated *t*-test was conducted using the limma R package (*Ritchie et al., 2015*; *Schwämmle et al., 2013*). Benjamini-Hochberg FDR-adjusted p values (q values) were used to control the false discovery rate.

## Antibodies

Antibodies for immunoblot and flow cytometry are detailed in the Key resources table. Anti-HIV-1 Vif (*Simon et al., 1995*) was obtained from the AIDS Reagent Program, Division of AIDS, NIAID, NIH (Dr M. H. Malim).

## Flow cytometry

### Antibody staining

For the flow cytometric screen in 293Ts, a sub-confluent 24-well/condition was harvested with trypsin-EDTA, fixed and permeabilised using the Cytofix/Cytoperm Fixation and Permeabilisation Kit (BD Biosciences) according to the manufacturer's instructions. Permeabilised cells were stained with

AF647-conjugated rabbit anti-HA antibody (Abcam) for 20 min at RT, washed, and analysed with an LSR Fortessa flow cytometer (BD Biosciences). Doublets were excluded by comparing SSC-W with SSC-H. Depletion of HA-tagged PPP2R5B, APOBEC3F, or APOBEC3G was quantified by the ratio of median AF647 fluorescence in GFP+ (transfected, Vif+)/GFP- (untransfected, Vif-) cells for each condition, after deducting background fluorescence of control 293Ts (no HA-tagged protein expression).

For titration of HIV-AFMACS viruses, typically $2 \times 10^5$ washed CEM-T4s were stained with fluorochrome-conjugated anti-LNGFR and anti-CD4 for 15 min at 4°C then fixed in PBS/1% paraformaldehyde and analysed as above. For titration of lentivectors, GFP fluorescence was quantified without antibody staining.

## DNA content

For cell cycle analysis in transduced CEM-T4s and transduced/transfected HeLas, $1 \times 10^6$ cells/condition (CEM-T4s) or a 50% confluent 6-well/condition (HeLas) were washed with PBS, then fixed for 30 min with ice-cold 90% methanol. Fixed cells were stained with 7-AAD at 25 µg/ml for 30 mins at 37°C, then analysed with an LSR Fortessa flow cytometer (BD Biosciences). Doublets were excluded by comparing SSC-W with SSC-H. The FlowJo cell cycle platform was used to determine the fraction of cells in each phase of cell cycle. G2/M cell cycle arrest was quantified by the ratio of cells in G2/M for each condition, compared with mock-transduced/transfected cells.

For cell cycle analysis in CEM-T4s infected with HIV-AFMACS, cells were first stained with FITC-conjugated anti-LNGFR (BioLegend), then washed, fixed and stained with 7-AAD and analysed as above. G2/M cell cycle arrest was quantified by the ratio of cells in G2/M for LNGFR+ (infected, HIV +)/LNGFR- (uninfected) cells for each condition.

For cell cycle analysis in primary human CD4+ T cells infected with HIV-AFMACS, CD3/CD28 Dynabeads were first removed using a DynaMag-2 magnet (Invitrogen). Cells were stained with FITC-conjugated anti-LNGFR (BioLegend) and BV421-conjugated anti CD4 (BioLegend), then washed, fixed, stained with 7-AAD and analysed as above. G2/M cell cycle arrest was quantified by the ratio of cells in G2/M for LNGFR+ CD4- (infected, HIV+)/LNGFR- CD4+ (uninfected) cells for each condition.

## Immunoblotting

Washed cell pellets were lysed in PBS/2% SDS supplemented with Halt Protease and Phosphatase Inhibitor Cocktail (Thermo Scientific) and benzonase (Sigma) for 10 min at RT. Post-nuclear supernatants were heated in Laemelli Loading Buffer for 5 min at 95 °C, separated by SDS-PAGE and transferred to Immobilon-P membrane (Millipore). Membranes were blocked in PBS/5% non-fat dried milk (Marvel)/0.2% Tween and probed with the indicated primary antibody overnight at 4 °C. Reactive bands were visualised using HRP-conjugated secondary antibodies and SuperSignal West Pico or Dura chemiluminescent substrates (Thermo Scientific). Typically 10–20 µg total protein was loaded per lane.

## Quantitative reverse transcription PCR

For quantification of PPP2R5 subunit mRNA levels, total RNA was extracted using TRIzol reagent (Invitrogen), followed by DNase I treatment. cDNA was synthesised using the Superscript III First-Strand Synthesis System (Invitrogen) with Oligo(dT) (Invitrogen), and 10 ng/test subjected to real-time PCR using the primers tabulated in *Supplementary file 1* (PCR primers for real-time PCR) and the SYBR Green PCR Master Mix (Thermo Scientific).

To control for PCR efficiency and allow comparison of abundances between different PPP2R5 subunits, standard curves were generated using plasmids encoding PPP2R5A, PPP2R5B, PPP2R5C, PPP2R5D and PPP2R5E (all Addgene), together with housekeeping gene Tata Binding Protein (TBP, Source Bioscience). Abundances of different PPP2R5 subunits for each cell type or condition were derived from the corresponding standard curves, and expressed as mRNA copy numbers relative to TBP as an endogenous reference.

### Visualization of Vif-CUL5 crystal structure

The previously determined structure of Vif in complex with CUL5, CBFβ, and ELOB/C (PDB ID: 4N9F) was used to identify solvent-exposed residues to be mutated in this study (*Guo et al., 2014*). Structural analysis and figures were generated using PyMOL Molecular Graphics System, Version 2.0 (Schrödinger).

### Bioinformatic analysis of Vif polymorphisms

Protein sequence Web Alignments for Vif, Env, Gag and Nef were downloaded from the Los Alamos HIV Sequence Alignments Database (accessible at: http://www.hiv.lanl.gov/). The following server options were selected: Alignment type, Web (all complete sequences); Organism, HIV-1/SIVcpz; Subtype, M group without recombinants (A-K); DNA/Protein, protein; Year, 2018; Format, FASTA.

These alignments contain all non-recombinant HIV-1 M group sequences from the Los Alamos HIV Sequence Database, with the following exceptions: only one sequence per patient is included; a single representative is included of very similar sequences; and sequences unlikely to represent naturally-occurring, viable viruses are excluded. We further subdivided the sequences according to viral clade (subtype) using the information in the sequence name for example B.FR.83.HXB2 is assigned to clade B. Analyses were conducted for both clade B viruses and all non-recombinant HIV-1 M group viruses.

To examine amino acid polymorphism in naturally occurring Vif variants at positions corresponding to residues 31, 33/34 and 128 of NL4-3 Vif, sequence logos were generated using WebLogo (*Crooks et al., 2004*). Further data analysis was conducted in R v.3.6.1 (*R Development Core Team, 2019*). In brief, residues at each position of interest were extracted using the seqinr R package (*Charif and Lobry, 2007*), then frequencies were calculated and graphical summaries generated using the ggplot R package (*Wickham, 2009*). To identify covariance (non-random association) between polymorphisms at positions 31 and 128, $2 \times 2$ contingency tables comparing frequencies of key residue pairs were constructed, then subjected to two-tailed Fisher's exact tests of independence (*Wang and Lee, 2007*).

To construct phylogenetic trees, only viruses with protein sequences available for all of Vif, Env, Gag and Nef were included. This enabled direct comparison of trees based on different viral proteins. Multiple sequence alignments and phylogenetic tree data (in Newick format) for each viral protein were generated using the Clustal Omega web server (*Sievers and Higgins, 2014*), then visualised using the ggtree R package (*Yu et al., 2018*).

All alignment and sequence files, scripts and details of the bioinformatic analyses described here are available at: https://github.com/annaprotasio/Marelli_et_al_HIV_Vif (copy archived at https://github.com/elifesciences-publications/Marelli_et_al_HIV_Vif).

### General statistical analysis

Where indicated, Student's *t*-tests (unpaired two-sample, assuming homoscedasticity, two-tailed), Fisher's exact tests (two-tailed) and 95% confidence intervals were calculated using Prism 7.0 (GraphPad). General data manipulation was conducted using Excel 2016 (Microsoft).

## Acknowledgements

This work was supported by the MRC (CSF MR/P008801/1 to NJM), NHSBT (WPA15-02 to NJM), the Wellcome Trust (PRF 210688/Z/18/Z to PJL), the NIHR Cambridge BRC, and a Wellcome Trust Strategic Award to CIMR. The authors thank Dr Reiner Schulte and the CIMR Flow Cytometry Core Facility team, and members of the Matheson and Lehner laboratories for critical discussion.

## Additional information

### Funding

| Funder | Grant reference number | Author |
|---|---|---|
| Medical Research Council | MR/P008801/1 | Nicholas J Matheson |
| NHS Blood and Transplant | WPA15-02 | Nicholas J Matheson |

| Wellcome | 210688/Z/18/Z | Paul J Lehner |

The funders had no role in study design, data collection and interpretation, or the decision to submit the work for publication.

## Author contributions

Sara Marelli, Conceptualization, Data curation, Formal analysis, Investigation, Visualization, Methodology; James C Williamson, Data curation, Formal analysis, Investigation, Methodology; Anna V Protasio, Data curation, Software, Formal analysis, Investigation, Visualization, Methodology; Adi Naamati, Investigation, Methodology; Edward JD Greenwood, Conceptualization, Resources, Methodology; Janet E Deane, Supervision; Paul J Lehner, Conceptualization, Resources, Supervision, Funding acquisition, Project administration; Nicholas J Matheson, Conceptualization, Data curation, Formal analysis, Supervision, Funding acquisition, Visualization, Methodology, Project administration

## Author ORCIDs

Edward JD Greenwood (iD) http://orcid.org/0000-0002-5224-0263
Janet E Deane (iD) http://orcid.org/0000-0002-4863-0330
Paul J Lehner (iD) http://orcid.org/0000-0001-9383-1054
Nicholas J Matheson (iD) https://orcid.org/0000-0002-3318-1851

## Ethics

Human subjects: Ethical permission for this study was granted by the University of Cambridge Human Biology Research Ethics Committee (HBREC.2017.20). Written informed consent was obtained from all volunteers prior to providing blood samples.

## Decision letter and Author response

Decision letter https://doi.org/10.7554/eLife.53036.sa1
Author response https://doi.org/10.7554/eLife.53036.sa2

# Additional files

## Supplementary files

• Supplementary file 1. DNA and RNA sequences. Sequences of PCR primers for Vif mutant library construction, codon-optimised Vif variants synthesised as gBlocks, Vif coding sequences in HIV-AFMACS viruses, the C-terminal 4xHA-tagged APOBEC3F coding sequence in pHRSIN-S-W-pGK puro, oligonucleotides for RNAi and primers for qRT-PCR.

• Transparent reporting form

## Data availability

All data generated or analysed during this study are included in the manuscript and supporting files. The mass spectrometry proteomics data have been deposited to the ProteomeXchange consortium via the PRIDE partner repository with the dataset identifier PXD018271 and are summarised in Source data files for Figures 2, 6 and 7.

The following dataset was generated:

| Author(s) | Year | Dataset title | Dataset URL | Database and Identifier |
|---|---|---|---|---|
| Marelli S, Williamson JC, Protasio AV, Naamati A, Greenwood EJD, Deane JE, Lehner PJ, Matheson NJ | 2020 | Antagonism of PP2A is an independent and conserved function of HIV-1 Vif and causes cell cycle arrest | https://www.ebi.ac.uk/pride/archive/projects/PXD018271/ | ProteomeXchange, PXD018271 |

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
