## [Decision Letter]

**Acceptance summary:**

In addition to inducing degradation of APOBEC3G, Vif also triggers proteasomal degradation of PPP2R5 family regulatory subunits of the host phosphatase PP2A. In the present study, the authors identified mutations in Vif separating these activities and provide evidence that the degradation of PPP2R5 family members accounts for the cell cycle arrest caused by certain Vif proteins. Thus, the findings provide new insights on how HIV-1 manipulates its host cells.

**Decision letter after peer review:**

Thank you for submitting your article "Antagonism of PP2A is an independent and conserved function of HIV-1 Vif and causes cell cycle arrest" for consideration by *eLife*. Your article has been reviewed by two peer reviewers, and the evaluation has been overseen by Frank Kirchhoff as the Reviewing Editor and Päivi Ojala as the Senior Editor. The following individual involved in review of your submission have agreed to reveal their identity: Linda Chelico (Reviewer #1).

The reviewers have discussed the reviews with one another and the Reviewing Editor has drafted this decision to help you prepare a revised submission.

Summary:

In the present study, Matheson, Lehner and coworkers followed up on their previous work showing that, in addition to inducing degradation of APOBEC3G, Vif also triggers the proteasomal degradation of all PPP2R5 family regulatory subunits of the host phosphatase PP2A. Here, they identified mutations in Vif separating these activities and provide evidence that the degradation of PPP2R5 family members accounts for the cell cycle arrest caused by certain Vif proteins. They also show that efficient depletion of all PPP2R5 subunits seems required for the Vif-dependent cell cycle arrest, providing an explanation for why HIV-1 Vif variants that antagonize some but not all PPP2R5 subunits do not cause cell cycle arrest.

The study addresses an important issue and for most part the data are clearly presented and convincing. Limitations are that the significance of this Vif activity for HIV-1 replication in primary cells and in vivo remain elusive although the authors provide some evidence for selective pressure to maintain this Vif function. Altogether, the findings provide relevant insights into the molecular biology of HIV-1 but as outlined below some additional experiments should be performed to further increase its significance.

Essential revisions:

1) Figure 1 and Vif mutants. An immunoblot of the Vif mutants would assist in determining their stabilities by observing steady state levels. Later in manuscript there appears to be some differences in Vif mutant stabilities that should be addressed directly in Figure 1, rather than citing that they are not predicted to interfere with stability.

2) In Figure 1, only APOBEC3G is used as a control, but since Vif binds APOBEC3F and APOBEC3H in different regions, it would be more convincing if all 3 APOBECs were used to determine the extent of overlap. This is partially addressed in later figures and so showing the data in Figure 1 for the 3 main APOBEC3s that have anti-HIV activity would make the later data, which shows some overlap in activity for some APOBEC3s, less contradictory to this figure.

3) The rationale for the cutoff of different amino acids in Figure 1C is not clear; why was R15A not chosen? Why is only PPP2R5B examined by flow cytometry in Figure 1C when PPP2R5D is shown in immunoblots in Figure 1A and Figure 2A? Figure needs to be more consistent or have consistent reasoning for data shown.

4) Figure 4: Conclusion that all PPP2R5B subunits are required; yet HeLa cells don't express the PPP2R5B subunit (that was previously shown as important/most conserved function and is highlighted in Figure 1B). Can the authors transfect PPP2R5B to supplement the HeLa cells to confirm that all subunits need to be degraded to see effect? PPP2R5B is used as the major example in all the mechanistic data; but lacks the key functional evidence. In supplementary figure for Figure 4, there is more PPP2R5B level than the other PPP2R5s, yet authors state that the mRNA is low to not detectable by qRT-PCR. The levels and importance of PPP2R5B needs to be better tested/presented. Especially since a recent article, Salamango et al., Cell Reports, 2019 implicates only PPP2R5A, C, and D in cell cycle arrest.

---

## [Author Response]

Essential revisions:1) Figure 1 and Vif mutants. An immunoblot of the Vif mutants would assist in determining their stabilities by observing steady state levels. Later in manuscript there appears to be some differences in Vif mutant stabilities that should be addressed directly in Figure 1, rather than citing that they are not predicted to interfere with stability.

Thank you for the suggestion, we have included the requested immunoblot as an additional figure supplement (Figure 1—figure supplement 3), and altered the main text accordingly. In summary, when overexpressed in 293Ts, steady state levels of the Vif mutants we found to be defective for PPP2R5B depletion (highlighted in Figure 1C) were comparable to wildtype.

Results:

“We discovered several Vif mutants to be defective for PPP2R5B depletion (Figure 1C). Vif protein expression levels were similar (Figure 1—figure supplement 3)…”

Please also note that, the aim of our flow cytometric screen was to identify Vif variants which lacked the ability to deplete PPP2R5B, but retained the ability to deplete APOBEC3G. A functional test of Vif stability was therefore built into the screen i.e. “stable enough to deplete APOBEC3G”.

This is important, because it is a direct test of the biologically relevant endpoint, controls for changes in recruitment of the CUL5 E3 ligase complex as well as steady state Vif levels, and avoids potential confounding by changes to anti-Vif antibody binding affinity caused by the point mutations. We make this point in the text.

Results:

“As well as indicating preserved APOBEC3 family substrate recruitment, the ability to deplete APOBEC3G served as a control for unanticipated effects on Vif expression or stability, or assembly of the Vif-CUL5 complex.”

We would like to take the opportunity to emphasise that, we have not observed any differences in Vif stability for the critical Vif I31 and I128 mutants which form the focus of this manuscript, based on either immunoblot or proteomic measurements of steady state levels, or flow cytometric or proteomic assays of APOBEC3 family depletion.

Conversely, as described in the Results and illustrated in Figure 2, mutations in R33/K34 were typically associated with somewhat lower Vif levels in CEM-T4s, accompanied by partial loss of activity against APOBEC3 family members, particularly APOBEC3F. This is why we chose to focus instead on residues 31 and 128.

2) In Figure 1, only APOBEC3G is used as a control, but since Vif binds APOBEC3F and APOBEC3H in different regions, it would be more convincing if all 3 APOBECs were used to determine the extent of overlap. This is partially addressed in later figures and so showing the data in Figure 1 for the 3 main APOBEC3s that have anti-HIV activity would make the later data, which shows some overlap in activity for some APOBEC3s, less contradictory to this figure.

Thank you for the suggestion, we have included the requested flow cytometric screen against APOBEC3F as an additional figure supplement (Figure 1—figure supplement 4), and altered the main text accordingly. In summary, APOBEC3F depletion by the Vif mutants we found to be defective for PPP2R5B (highlighted in Figure 1C) was preserved, with the exception of the R15A mutant (as previously reported) and the RK33/34AA mutant.

Results:

“First, we tested the ability of Vif mutants lacking the ability to deplete PPP2R5B to deplete HA-tagged APOBEC3F in 293Ts, similar to our initial flow cytometry screen (Figure 1—figure supplement 4A-C). As previously reported (Letko et al., 2015; Nakashima et al., 2016), mutation of R15 resulted in loss of activity against APOBEC3F. The RK33/34AA mutant was also partially impaired, but other mutants retained full activity against APOBEC3F.”

Please note that, these results concord precisely with our proteomic data (shown in Figure 2D). In fact, a major strength of our study is the use of a functional proteomic approach to assess Vif activity against endogenous APOBEC3 and PPP2R5 family members in T cells, avoiding the possibility of over-expression artefacts, or the need for protein tags.

Since APOBEC3F was not quantitated in our second proteomic experiment (shown in Figure 6A-C), however, we took the opportunity to use the APOBEC3F flow cytometric screen to formally test the activity of our critical “Vif AYR” double mutant (combined I31/I128 mutations). This data is now included in an expanded figure supplement (Figure 6—figure supplement 3A-B, lower panels), and we have altered the main text accordingly. In summary, and as predicted, Vif AYR retained activity against APOBEC3F.

Results:

“Since PPP2R5B and APOBEC3F were not quantitated in the proteomic experiment, we tested the ability of Vif AYR to deplete these subunits by flow cytometry in 293Ts, including APOBE3G as a control (Figure 6—figure supplement 3A-B). As expected, Vif AYR was able to deplete APOBEC3F and APOBEC3G but not PPP2R5B, and failed to cause cell cycle arrest (Figure 6D).”

Please also note that, we did not screen against APOBEC3H haplotype II, because wildtype NL4-3 Vif (on which our mutant library was based) is known to be unable to deplete this APOBEC3 family member. For clarity, we now include a better overview of the AOBEC3 family, and explicitly make this point in the main text.

Results:

“As well as APOBEC3G, other APOBEC3 family members (such as APOBEC3F and APOBEC3H haplotype II) are also able to restrict HIV replication (Feng et al., 2014), and Vif recruits different APOBEC3 family members for degradation using distinct binding surfaces (Binka et al., 2012b; Chen et al., 2009; Dang et al., 2009; Gaddis et al., 2003; Harris and Anderson, 2016; He et al., 2008; Letko et al., 2015; Mehle et al., 2007; Nakashima et al., 2016; Ooms et al., 2016; Richards et al., 2015; Russell and Pathak, 2007; Simon et al., 2005a; Yamashita et al., 2008).”

Results:

“APOBEC3H haplotype II was not examined, because wildtype NL4-3 Vif (on which our Vif mutant library was based) is unable to deplete this APOBEC3 family member (Binka et al., 2012a; Ooms et al., 2013; Zhao et al., 2015).”

3) The rationale for the cutoff of different amino acids in Figure 1C is not clear; why was R15A not chosen? Why is only PPP2R5B examined by flow cytometry in Figure 1C when PPP2R5D is shown in immunoblots in Figure 1A and Figure 2A? Figure needs to be more consistent or have consistent reasoning for data shown.

Thank you for the comment/questions, we are sorry that these points were not clear to the reviewer. In brief: first, we did not evaluate the R15A mutant further because it was already known to be defective for APOBEC3F depletion (now confirmed here in Figure 1—figure supplement 4), and we were seeking Vif mutants which lacked the ability to deplete PPP2R5 family subunits, but retained the ability to deplete APOBEC3 family members.

Second, we chose PPP2R5B as a representative PPP2R5 subunit for our flow cytometric screen because we had previously shown that, amongst the PPP2R5 family subunits, depletion of PPP25B was most conserved across Vif variants from different primate lentiviruses.

Third, we chose PPP2R5D as a representative PPP2R5 subunit for our immunoblots because, to our knowledge, no validated PPP2R5B antibody is available, and, amongst the other PPP2R5 family subunits, this is (in our experience) the most reliable antibody reagent.

To ensure that our reasoning is clear, we have reviewed and in some cases rewritten the main text to emphasise the overall strategy, and explicitly address these points.

Results:

“We discovered several Vif mutants to be defective for PPP2R5B depletion (Figure 1C). Vif protein expression levels were similar (Figure 1—figure supplement 3), but some mutations affected residues already known to be required for depletion of APOBEC3G (K26, Y44, W70) (Letko et al., 2015) or APOBEC3C/F (R15) (Letko et al., 2015; Nakashima et al., 2016) (Figure 1D and Figure 1—figure supplement 2C). Conversely, Vif variants with mutations in residues Y30/I31, R33/K34 and I128 were defective for PPP2R5B depletion, retained the ability to antagonise APOBEC3G, and had not been implicated in APOBEC3C/F depletion. These residues are grouped in three similarly orientated patches on the Vif surface (Figure 1B, residues highlighted in red). Aiming to identify Vif variants specifically defective for PPP2R5 subunit depletion, we therefore focused on mutations in residues I128, I31 and R33/K34 for further evaluation, including representatives from each patch.”

Results:

“Amongst the five PPP2R5 family subunits, we previously showed that depletion of PPP2R5B is most conserved across Vif variants from HIV-1/2 and the non-human primate lentiviruses (Greenwood et al., 2016). We therefore transfected our library into HEK 293T cells (293Ts) stably expressing HA-tagged PPP2R5B or APOBEC3G, and used flow cytometry to quantify PPP2R5B and APOBEC3G depletion by each Vif variant…”

Results:

“Levels of an indicative PPP2R5 subunit for which a reliable antibody is available (PPP2R5D, as in Figure 1A) were then measured by immunoblot (Figure 2A).”

4) Figure 4: Conclusion that all PPP2R5B subunits are required; yet HeLa cells don't express the PPP2R5B subunit (that was previously shown as important/most conserved function and is highlighted in Figure 1B). Can the authors transfect PPP2R5B to supplement the HeLa cells to confirm that all subunits need to be degraded to see effect? PPP2R5B is used as the major example in all the mechanistic data; but lacks the key functional evidence. In supplementary figure for Figure 4, there is more PPP2R5B level than the other PPP2R5s, yet authors state that the mRNA is low to not detectable by qRT-PCR. The levels and importance of PPP2R5B needs to be better tested/presented. Especially since a recent article, Salamango et al., Cell Reports, 2019 implicates only PPP2R5A, C, and D in cell cycle arrest.

Thank you for the suggestions, we have included the requested PPP2R5B over-expression and qRT-PCR experiments as an additional figure supplement (Figure 4—figure supplement 2), added a further figure supplement examining Vif-dependent cell cycle arrest in primary human CD4^+^ T cells (Figure 7—figure supplement 2), and altered the main text accordingly. These revisions are described in more detail below.

First, to control for PCR efficiency and allow formal comparison of abundances between different PPP2R5 subunits, we have repeated all our qRT-PCR data using standard curves generated from plasmids encoding PPP2R5A-E and TBP. Copies of mRNA (normalised to TBP) for PPP2R5A-E in CEM-T4s and HeLas are now shown in Figure 4—figure supplement 2A. In summary, these data confirm that expression of PPP2R5B is detectable in both cell types, but lower than other PPP2R5 subunits, particularly in HeLas.

Results:

“First, we confirmed that, as in CEM-T4s, expression of WT NL4-3 Vif in HeLas causes cell cycle arrest (Figure 4—figure supplement 1B). Likewise, mRNA expression levels of individual PPP2R5 subunits were determined by quantitative real-time PCR (qRT-PCR), and found to be similar between cell types, with PPP2R5B much lower than other subunits (Figure 4—figure supplement 2A).”

Please note that, we have also repeated our qRT-PCR test of siRNA knockdown efficiency for each subunit, using the same method (Figure 4—figure supplement 2B). In this figure, relative mRNA abundance for each subunit is shown for siRNA-transfected vs control cells. Because expression of PPP2R5B in wildtype cells is lower than other subunits, expression falls below the linear range of the assay in cells transfected with siRNA.

Second, we over-expressed PPP2R5B in HeLas, resulting in elevated PPP2R5B mRNA levels comparable to other subunits (Figure 4—figure supplement 2C). As predicted by the reviewer, supplementation with PPP2R5B rescued cell cycle progression in the presence of PPP2R5A, C, D and E depletion by siRNA (Figure 4—figure supplement 2C-E).

Results:

“That depletion of PPP2R5B is neither sufficient (Figure 4A-B) nor required (Figure 4C-D) in this setting may reflect low expression of PPP2R5B (Figure 4—figure supplement 2A), consistent with previous protein-level data from HeLas (Geiger et al., 2012). To test this hypothesis, we generated HeLa cells expressing exogenous PPP2R5B at similar levels to other PPP2R5 subunits (Figure 4—figure supplement 2C). As predicted, transfection of a pool of siRNA targeting PPP2R5A, C D and E (but not B) caused cell cycle arrest in wildtype HeLas, but not HeLas over-expressing PPP2R5B (Figure 4—figure supplement 2D-E, compare with Figure 4C-D).”

Please also note the reasons described above for focussing on PPP2R5B as a representative PPP2R5 subunit for our flow cytometric screen. In fact, our data suggest that efficient depletion of all PPP2R5 subunits is required to cause cell cycle arrest. Critically, this explains why Vif variants with impaired activity against any PPP2R5 subunit (rather than one particular PPP2R5 subunit) are defective for this phenotype (compare the differential effects of Vif variants with mutations in I31 and I128 shown in Figure 2C). We elaborate on this in our Discussion (paragraph three).

Finally, to confirm that the effects we describe in CEM-T4s and HeLas are relevant to HIV-1 infection of primary human CD4^+^ T cells, we infected cells from 2 donors with WT, ΔVif or Vif AYR viruses (on a ΔVpr background), and confirmed that the combined I31/I128 mutations in Vif AYR did indeed abolish Vif-dependent cell cycle arrest (Figure 7—figure supplement 2).

Results:

“To confirm a functional effect on PP2A, we then used these viruses to infect CEM-T4 T cells, and measured their effect on cell cycle progression. Again, only ΔVpr-Vif WT virus, but not ΔVpr-ΔVif or ΔVpr-Vif AYR viruses, was able to induce G2/M cell cycle arrest (Figure 7C). A similar effect was observed in HIV-infected primary human CD4^+^ T cells (Figure 7—figure supplement 2A-B).”